# Thermal dissociation cavity ring-down spectrometer (TD-CRDS) for detection of organic nitrates in gas and particle phase

**Natalie I. Keehan,[1] Bellamy Brownwood,[1] Andrey Marsavin,[1] Douglas A. Day,[2] and Juliane L. Fry[1,*]**

[1]Chemistry Department and Environmental Studies Program, Reed College, Portland, OR 97205, USA

[2]Cooperative Institute for Research in Environmental Sciences (CIRES) and Department of Chemistry, University of Colorado, Boulder, CO 80303, USA

*Correspondence to*: J. L. Fry (fry@reed.edu)

**Abstract.** A thermal dissociation – cavity ring-down spectrometer (TD-CRDS) was developed to measure $NO_2$, peroxy nitrates (PNs), alkyl nitrates (ANs), and $HNO_3$ in the gas and particle phase, built using a commercial Los Gatos Research $NO_2$ analyzer. The detection limit of the TD-CRDS is 0.66 ppb for ANs, PNs, and $HNO_3$ and 0.48 ppb for $NO_2$. For all four classes of $NO_y$, the time resolution for separate gas and particle measurements is 8 minutes and for total gas + particle measurements is 3 minutes. The accuracy of the TD-CRDS was tested by comparison of $NO_2$ measurements with a chemiluminescent NOx monitor, and aerosol-phase ANs with an Aerosol Mass Spectrometer (AMS). $N_2O_5$ causes significant interference in the PNs and ANs channel under high oxidant concentration chamber conditions, and ozone pyrolysis causes a negative interference in the $HNO_3$ channel. Both interferences can be quantified and corrected for, but must be considered when using TD techniques for measurements of organic nitrates. This instrument has been successfully deployed for chamber measurements at widely varying concentrations, as well as ambient measurements of $NO_y$.

## 1 Introduction

Nitrogen oxide based functional groups are an area of significant interest in atmospheric oxidative chemistry. Organic nitrates are formed through reactions between volatile organic compounds (VOCs), of which the global majority are biogenic in origin (Seinfeld and Pankow 2003; Perring, Pusede, and Cohen 2013) and NOx (=NO+$NO_2$) or $NO_3$ (Ng et al. 2017), which is predominantly anthropogenic in origin (Seinfeld and Pandis 2006). The two major organic nitrate products of these reactions are alkyl nitrates (ANs) of the form $RONO_2$ and peroxy nitrates (PNs) of the form $ROONO_2$. These organic nitrates play an important role in regulating ozone in the troposphere by serving as temporary reservoirs of $NO_2$ (Buhr et al. 1990; Thornton et al. 2002). Equilibrium partitioning of high molecular weight, low volatility organic molecules occurs, causing some organics to condense onto existing particles (Jimenez et al. 2009). These secondary organic aerosols (SOA) consist primarily of the highly oxidized products of VOC + oxidant reactions, because of their increased molecular weight and higher polarity. Lower night-time temperatures decrease volatility even further, leading to increased partitioning into the particle phase (Fry et al. 2013). Warmer temperatures, deposition, and chemistry within the particles change the equilibrium, resulting in the release of $NO_2$. Because of long residence times of SOA, significant quantities of $NO_2$ can be transported away from source regions by

wind in reservoir form (Perring, Pusede, and Cohen 2013; Browne et al. 2013; Wolfe et al. 2007; Kim et al. 2014; Zare et al. 2018).

Different classes of organic nitrates dissociate in distinct temperature ranges, based upon the inherent stability of the molecules. At residence times of 30-90 ms in quartz tubes, peroxy nitrates (PNs, $RO_2NO_2$) dissociate at approximately 150ºC, alkyl nitrates (ANs, $RONO_2$) at 350ºC, and nitric acid ($HNO_3$) at 600ºC (Day et al. 2002). The dissociation temperatures are dependent on residence times, but there seems to be very little dependence on what constitutes the R group (Hao et al. 1994; Kirchner et al. 1999). This is useful for the detection of total peroxy and alkyl nitrates ($\Sigma$PNs and $\Sigma$ANs, respectively) because they can be dissociated as a class, with identical detection efficiency regardless of the chemical nature of the R group. Reaction 1 shows that the thermal dissociation of each class of organic nitrates results in one $NO_2$ and a hydrocarbon-containing X group.

$$XNO_2 + \Delta \rightarrow X + NO_2 \qquad\qquad (R1)$$

where X = $RO_2$, RC(O)OO, RO, or OH.

PNs serve as a temporary reservoir of $NO_2$ in the atmosphere, because the equilibrium between formation and dissociation is rapid. For example:

$$CH_3O_2 + NO_2 \leftrightarrow CH_3O_2NO_2 \qquad\qquad (R2)$$

has a $K_{eq}$ of 2.2 x $10^{-12}$ $cm^3$ molecules$^{-1}$, resulting in a PN lifetime at 20 ppb $NO_2$ of 0.56 seconds at 298K and 1 bar (Atkinson et al. 2006; JPL Data Evaluation 2015). In contrast, ANs and $HNO_3$ predominantly serve as sinks of $NO_2$, with spatial transport scales that depend on their meteorology-dependent deposition lifetimes (Horowitz et al. 2007).

Previous studies of organic nitrates have been done by measuring specific nitrates (Wolfe et al. 2007; Horowitz et al. 2007; Parrish and Fehsenfeld 2000; Surratt et al. 2006; Lee et al. 2016) or by looking at the sum of nitrates using thermal dissociation $NO_2$ measurements (Zellweger et al. 1999; Day et al. 2002; Hargrove and Zhang 2008; Paul, Furgeson, and Osthoff 2009; Rollins et al. 2010; Sobanski et al. 2016). The instrument described in this paper has drawn on aspects of three different thermal dissociation nitrate measurement strategies in the literature. The general oven and flow plan was based on the thermal dissociation-laser induced fluorescence (TD-LIF) instrument built by the Cohen group at UC Berkeley (Day et al. 2002). Instead of LIF, the $NO_2$ detection device in the instrument described here is a commercial cavity ring-down spectrometer (CRDS). Once interferences are characterized and absorption cross-sections are known, CRDS does not require in-line calibration by an authentic standard gas cylinder during sample measurement, as discussed in Paul et al. (Paul, Furgeson, and Osthoff 2009). Gas-particle partitioning measurements using a switchable charcoal denuder was incorporated from Rollins et al. (Rollins et al. 2010).

The benefit of using CRDS over chemiluminescence (CL) detection of $NO_2$ is its selectivity. The (partial) thermal dissociation of multiple unstable nitrate compounds like ANs, PNs, and $N_2O_5$ into $NO_2$ by the CL heating process and molybdenum catalyst has been well documented (Wooldridge et al. 2010). CRDS can make direct measurements of $NO_2$, unlike CL, which uses a metal catalyst to turn $NO_2$ into NO and back-calculates $NO_2$ concentration by subtraction. CRDS does not require heating or a catalyst, and is therefore more selective. LIF can be tuned to a specific spectroscopic transition like CRDS,  and can be run

at lower cell pressures that reduce recombination (see section 3.7 below), but laser power becomes limiting for measurement of low concentrations, and requires delicately aligned multipass optical cells to achieve low limits of detection for $NO_2$. The downsides of CRDS come from the expense and delicateness of the instrument.

Since high molecular weight oxidation products can condense into the particle phase, it is valuable to be able to make both gas and particle phase measurements. Denuders work by using diffusion to separate gases from liquid- or solid-phase particles. Higher diffusion rates for gases means that they are more readily absorbed into the walls of a charcoal denuder, leaving behind the particle phase. The fraction of gas removed depends on residence time in the denuder and the surface area available to diffusing gas molecules. The diffusion coefficient of $NO_2$ is reported to be 0.154 $cm^2s^{-1}$ (Williams et al. 2012) and 0.070 $cm^2s^{-1}$ for n-propyl nitrate (Paul, Furgeson, and Osthoff 2009). According to previous studies using charcoal denuders, the denuder removed the majority of particles with diameters <0.1 μm (Glasius et al. 1999) as well as all semivolatile organic gases.

## 2 Instrument design

In order to measure concentrations of organic nitrates by thermal dissociation, a multi-channel, switchable, controllable heating inlet system was constructed. This heating unit was then attached to a cavity ring-down $NO_2$ detector (CRDS, Los Gatos Research Inc. Model #907-0009) to complete the instrument. An overall instrument schematic is shown in Figure 1a.

The three quartz tube ovens were constructed out of 55 cm long, 3.8 mm inner diameter (ID), 7.0 mm outer diameter (OD) quartz tube wrapped in 15 cm nichrome wire (2mm wide ribbon with resistivity of 11 Ω/m) located 5 cm from one end. An 18 cm long, 8 mm ID, 10 mm OD quartz tube was slipped around the nichrome section to hold it in place. Over the 8mm ID tubing, two 8 cm long, 10.5 mm ID, 13 mm OD quartz tubes were placed with a thermocouple in between to hold the end of the thermocouple in place. The whole heated section was wrapped in ½" thick ceramic insulation (McMaster-Carr # 9379K92) with foil coating, as shown in Figure 1b. It is important to note that the heat capacity of the oven is determined by the effectiveness of the insulation. Insufficient insulation can result in unstable oven temperatures or increase the time required for the gas to reach the required dissociation temperature, leading to increased sampling times on each oven and a degradation of the time resolution of the instrument.

The thermocouple was placed so it was the same distance and glass thickness from the nichrome as the nichrome was from the gas flow, so it was hypothesized that the thermocouple temperature reading would be representative of the internal temperature of the oven. Experiments comparing this external thermocouple to a thermocouple placed at the same position inside the gas flow showed that the average internal oven temperature was between 25 and 30°C hotter than the external thermocouple reading. Because of the oven design, the temperature inside the heated portion of the oven is not uniform, but is hottest closest to the end of the nichrome section, nearer to the exhaust.

 The unheated portion of the quartz tubing used in this instrument is significantly shorter in length than the length originally calculated in Day et al. (2002) due to additional testing reported in Paul et al. (2009) (Paul, Furgeson, and Osthoff 2009). The shorter length was chosen to suppress the recombination reaction of $NO_2$ radical with the organic sister product upon cooling.

The shorter ovens were shown to effectively reduce residence time, and therefore recombination, but still allowed adequate time for gas cooling before entering the sampling chamber. A length of 55 cm was calculated from Equation 1 using the Paul et al. CRDS flow rate (q = 2.5 lpm) and oven length (h= 64 cm) in order to give our instrument equal residence times ($\tau$, see Eq. 1). Since the flow rate of the LGR CRDS is significantly smaller (1.2 lpm), the required tube length is shorter than that reported in Paul et al.

$$\tau = \frac{\pi r^2 h}{q}$$ (Eq. 1)

These ovens were attached to nominal ¼ inch (0.635 cm) Teflon tubing with Teflon Swagelok tees and unions. Teflon connectors were chosen over stainless steel to reduce destruction of $NO_2$ by heated steel (Hargrove and Zhang 2008). An oven-length piece of ¼ inch (0.635 cm) Teflon is used as the ambient temperature background $NO_2$ channel, which has a typical temperature of 22 - 24º C inside the inlet box. The three ovens and background channel connect to a six-port solenoid valve

with Teflon wetted surfaces. The outlet of the solenoid valve runs to the inlet of the LGR CRDS.

The inlet of the instrument has two possible pre-oven pathways: denuded and undenuded. The denuder is a 45 cm long cylinder of activated charcoal with a ¼ inch (0.635 cm) channel through the center. A three-way, Teflon-wetted solenoid directs the inlet air either through the denuder or through an equivalent length of Teflon tubing before the air sample enters the ovens.

An Omega CN616TC1 Temperature Controller was used to regulate the temperature of the ovens. The inlet end of the oven

nichrome wire was attached to the positive terminal of the Mouser 24VDC power supply and the exhaust end was wired to a Mouser DR06D12 solid state relay. These relays received signals from the temperature controller, either allowing or prohibiting current flow through the nichrome wire by completing the circuit loop. The temperature controller was able to detect the temperature of the ovens using K-type thermocouples. The desired temperatures were set using the CN616 Software provided with the temperature controller. Experiments showed that a single 24V power supply did not provide enough current

to heat the Channel 1 oven to an appropriate temperature, so a second 24V power supply was used to supply power to Channel 1. This succeeded in getting the oven as high as 820ºC; the typical temperature setpoint was 700ºC.

Valve switching was controlled by a Measurement Computing (MCC) USB-ERB08 relay module. Each solenoid was soldered to a diode to prevent damage from voltage spikes generated by switching. These leads were then connected to the normally closed (NC) ports of the MCC relay unit, which completed the circuit to open the specified valve.

One limitation of the TDCRDS instrument is its reliance on a single detector. This necessitates sequential measurements of each relevant species, creating a minimum time resolution for the instrument. This minimum time resolution can be large compared to the rate of change of the measured species in the atmosphere or in a chamber experiment. Any concentration changes faster than the timescale of the channel cycle are accounted for by assuming a linear change in each channel between two consecutive samplings of that channel, and using the interpolated values at the timescale of the measuring channel for

subtractions. This simplifying assumption only holds if the time between channel samplings is relatively short, and if there are no changes in background $NO_2$ on the timescale of the oven cycling. In situations where rapid $NO_2$ changes are likely, a parallel fast time resolution $NO_2$ measurement could be used to enable corrections for changing $NO_2$ background. The goal is

to minimize the instrument time resolution by minimizing the sampling time of each oven without introducing error caused by mixing analyte in the tubing between the switching valve and the CRDS sample cell. Plausible channel switching rates between

30 and 90 seconds were tested to measure the stabilization time of each channel. This testing was conducted by flowing 10 sccm of zero air through a three-necked round bottom flask containing 0.2 ml of isobutyl nitrate (IBN) chilled to -21°C. This 10 sccm flow was diluted with 7.25 lpm of zero air to achieve a concentration of ~ 700 ppb.

Figure S2 shows the $NO_2$ vs time curve for the TD-CRDS when gas was sampled with various lengths of time in each oven. The high concentration peaks are when the IBN dilution is flowing through Channels 1 and 2 (650°C and 385°C, respectively)

and the troughs are IBN flowing through Channels 3 and 4 (120°C and ambient 23°C, respectively). For this application, 45-second channel time yielded the best trade-off between channel stabilization and time resolution. (We note that maintaining a constant flow through each of the channels at all times would help to reduce the stabilization time in the CRDS, leading to a reduced time resolution. Because the CRDS has its own internal pump to draw air into the CRDS cell, a secondary pump would be required to maintain constant air flow through the non-sampling channels. Such a modification could help make this

instrument more viable for high time resolution ambient measurements.)

Channel timing with the denuder was determined in a similar manner, and it was determined that 1 minute per channel was necessary to achieve stabilization with the charcoal denuder. This leads to an 8 minute complete cycle time, since there is 1 minute for denuded and 1 minutes undenuded on each of the four species channels. The last 3 measured points in each channel period are averaged to obtain the concentrations that are used for each channel. Since the CRDS-$NO_2$ takes a measurement

every 1 sec, the last three measured points represent 3 seconds of sampling time. A full cycle in this gas / aerosol mode is shown in Figure 2.

For each full cycle of concentration measurements from the eight channels, the concentrations of the individual classes of NOy are determined by subtractively as follows. Section 3.9 below discusses subsequent corrections that are applied to each channel.

$$\text{Total } NO_2 = [NO_2]_{\text{oven 4}} \qquad\qquad \text{Aerosol } NO_2 = [NO_2]_{\text{oven 4, \textbf{denuded}}} \qquad\qquad \text{(Eqs. 2)}$$

$$\text{Total PNs} = [NO_2]_{\text{oven 3}} - [NO_2]_{\text{oven 4}} \qquad \text{Aerosol PNs} = [NO_2]_{\text{oven 3, \textbf{den}}} - [NO_2]_{\text{oven 4, \textbf{den}}}$$

$$\text{Total ANs} = [NO_2]_{\text{oven 2}} - [NO_2]_{\text{oven 3}} \qquad \text{Aerosol ANs} = [NO_2]_{\text{oven 2, \textbf{den}}} - [NO_2]_{\text{oven 3, \textbf{den}}}$$

$$\text{Total HNO}_3 = [NO_2]_{\text{oven 1}} - [NO_2]_{\text{oven 2}} \qquad \text{Aerosol HNO}_3 = [NO_2]_{\text{oven 1, \textbf{den}}} - [NO_2]_{\text{oven 2, \textbf{den}}}$$

The "Total" concentrations in Eqs. 2 refer to gas + aerosol phase, to obtain gas-phase only concentrations, the aerosol can be subtracted from the total for each channel. While there is not expected to be any signal in the $NO_2$ aerosol channel, the channel

has proven useful for diagnosing contamination problems, interferences not yet accounted for, and false values caused by rapid changes.

## 3 Calibration and characterization

### 3.1 Determination of NO₂ sensitivity

Two sets of tests were performed to verify the sensitivity of the LGR CRDS to $NO_2$. Response at high concentrations was
verified at concentrations of 250 to 1000 ppb using dilutions of $NO_2$ in zero air. A 514.5 ppm calibrated mixture of $NO_2$ in $N_2$
(Airgas) was diluted with a zero air source to generate the required mixing ratios. Response at low concentrations was
compared to a Thermo chemiluminescent NOx detector between 1.5 to 11.5 ppb. Low concentration $NO_2$ was obtained using
ambient lab air diluted using zero air. The results of these experiments are shown in Figure 3.

The fit line has a slope of close to 1 over both measured ranges, indicating good agreement under both high and low
concentration conditions. Since this experiment was performed using dilutions of zero air, any interference from $NO_y$ species
in the CL detector would also be expected to scale with dilution. The urban location of the lab would support the relatively
low levels of $NO_y$ compared to $NO_2$, explaining the very small 4% difference in slope between the two detectors. These
experiments suggest an upper limit error due to the $NO_2$ detection of 10%. The intercept offset of the low concentration
experiment is 0.64 ppb, which may be attributable to the interference of organic nitrates in the chemiluminescence
measurement, or a slight zero offset in the chemiluminescence detector. Thus, the CRDS is accurate under both atmospherically
relevant and elevated laboratory experiment conditions, but regular calibration against a known source or comparison with
another $NO_2$ measurement is nevertheless recommended.

### 3.2 Production of alkyl nitrates and peroxy nitrates using the Reed Environmental Chamber (REC) for TD-CRDS instrument characterization

The 400 L Teflon bag Reed Environmental Chamber (REC, Draper et al. 2015) was used to generate VOC + $NO_3$ reaction
products that could be analyzed using the TD-CRDS. The REC chamber was operated with steady inlet flows to the top of the
chamber of zero air (4.3 lpm), $O_3$ (200 sccm), $NO_2$ (4.4 sccm of 515 ppm), and VOC (14.2 sccm zero air through chilled liquid
source containing gas-phase VOC of ~100 ppm), which mix and react (average residence time ~ 90 minutes) and are sampled
for analysis from the bottom of the chamber. Zero air was generated using a Sabio Model 1001 zero air generator, which
removes water, particulates, and reactive gases. Ozone was generated using a UV light source (Pen-Ray Hg lamp at 254 nm)
inside the middle neck of a three-necked round bottom flask, and the concentration was altered by adjusting the depth of the
light source in the flask. The constant $NO_2$ source was a gas cylinder (Airgas, concentration analyzed 4/17/2013) with a
concentration of 514.5 (± 2%) ppm $NO_2$ in $N_2$. Approximately 300 ppb VOCs (typical VOCs used are Δ-carene, limonene, α-
pinene or β-pinene) were generated by flowing zero air over a chilled liquid sample of VOC in a three-necked round bottom
flask.

Ozone, zero air, and $NO_2$ flows were allowed to stabilize inside the chamber prior to introducing VOC flow to initiate the
experiment. All flows were then continuous until the completion of the experiment. Particle number and size data was collected
using a Scanning Electron Mobility Sizing (SEMS, Brechtel Manufacturing, Inc.), connected via conductive silicone tubing to

minimize particle losses. Ozone concentration was measured using a Dasibi Model 1003-AH ozone monitor or Teledyne Model T400.

### 3.3 Determination of oven temperature setpoints

Temperature ramps were performed on different mixtures of known gases to determine the appropriate setpoint temperatures for each of the three ovens. Temperature ramp results were used to identify the correct setpoints for each oven to achieve complete dissociation for each species. Both $HNO_3$ and AN measurements were performed by flowing zero air over a pure liquid analyte sample. The AN standard used was isobutyl nitrate (Aldrich 96% purity). A pure liquid sample of PN could not be obtained, so a $NO_3$ + Δ-3-carene mixture containing PNs was synthesized in the chamber as described above. Because concentrations were very stable, ramps were performed at 5 °C/min. Normalized measured $NO_2$ concentrations are plotted against temperature in Figure 4 because absolute concentrations were different for each class of nitrate.

Complete dissociation of PNs occurred at a thermocouple temperature reading of 130 °C, ANs at 385 °C, and $HNO_3$ at 600°C. The $HNO_3$ oven setpoint was chosen to be 700°C to allow the quantification of interference from $NO_3$ dissociated from $N_2O_5$ in that channel. At 600°C, $HNO_3$ is completely dissociated, but there is only partial conversion of $NO_3$ to $NO_2$, creating an interference in the hot channel in the TD-CRDS. At 700°C, $HNO_3$ is completely dissociated and $NO_3$ is completely converted to $NO_2$. Note that the dissociation plateaus do not overlap with the beginning of the adjacent curve, confirming the ability to quantitatively separate nitrate species by temperature.

### 3.4 Quantification and treatment of $N_2O_5$ interference

High concentration Δ-3-carene nitrate oxidation experiments in the REC chamber typically had 650 ppb $O_3$ and 400 ppb $NO_2$. When high concentrations of $O_3$ and $NO_2$ are present, they react in the chamber to form $N_2O_5$. This was verified by performing a temperature ramp from a chamber at low and high oxidant concentrations (Figure 5). $N_2O_5$ dissociates to produce two $NO_2$ products (see R3 and R4 below) across a broad temperature range, in contrast to the sharp dissociation curves for peroxy- or alkyl- nitrates, such that the presence of $N_2O_5$ removes the clear plateau between PNs dissociation and ANs and give an interference in both PNs and ANs channels. A chamber with low $NO_x$ conditions (335 ppb $O_3$ and ~3 ppb $NO_2$) that was left to equilibrate for 56 minutes after addition of Δ-carene gave a maximum $N_2O_5$ concentration of 1.6 ppb. The resulting temperature ramp gives the expected dissociation curve, showing both PNs and ANs plateaus (Figure 5). In this case, there is good separation between PNs and ANs, because $N_2O_5$ is lower in concentration. This $N_2O_5$ interference has been previously observed by Womack et al. 2017. Note that given the gradual dissociation of $N_2O_5$ across this full temperature range, the extent of the interference depends on the exact temperature setpoints, so any similar TD-based organonitrate instrument that may be operated in high-$N_2O_5$ conditions should characterize its individual $N_2O_5$ interference.

To measure the $N_2O_5$ interference such that it can be corrected for, we ran an experiment with only oxidants in the chamber (Figure 6). The TD-CRDS detects one $NO_2$ molecule from the first dissociation of $NO_2$ from $N_2O_5$ (Reaction 3) either in Oven 3 (the PNs channel) or in Oven 2 (the ANs channel), and another $NO_2$ is observed when the released $NO_3$ fragment further

dissociates in the HNO₃ channel (Reaction 4). We note that due to its high reactivity and wall losses (especially the $NO_3$ fragment), as well as the likelihood that some $N_2O_5$ remained incompletely dissociated even at the ANs oven temperature, the total $N_2O_5$ detection is substantially less than 100% of the $N_2O_5$ concentration present in the chamber. We also emphasize that these percentages are specific to the configuration used in this characterization experiment: from the chamber containing the

modeled $N_2O_5$ concentration used to determine these interference percentages, a 2-m Teflon inlet line led to the TD-CRDS instrument. A kinetic model paired with measurements of $NO_2$ and $O_3$ in order to predict $N_2O_5$ can be used to quantify the interferences in each channel for a given setup.

$$N_2O_5 \rightarrow NO_2 + NO_3 \hspace{5cm} (R3)$$

$$NO_3 \rightarrow NO_2 + O \hspace{5.5cm} (R4)$$

The result of $N_2O_5$ is an elevated baseline in each of the PNs, ANs, and HNO₃ channels before the VOC is added. If an accurate $N_2O_5$ measurement is available, the interference from $N_2O_5$ (and $NO_3$) can be subtracted out of each channel, and these pre-VOC injection signals can be used to assess the likely lower inlet transmission of $N_2O_5$ and $NO_3$ vs. the more stable PNs, ANs, and HNO₃. In the absence of a separate $N_2O_5$ measurement, kinetic modeling can be used to predict how much $N_2O_5$ will be formed in each experiment, which can then be subtracted out. For example, in chamber experiments, comparing modeled $N_2O_5$

amounts to the amounts of signal in the ANs, PANs, and HNO₃ channels before VOC is added can quantify what fraction of $N_2O_5$ appears in each channel. If Reaction 3 happens across the PNs and ANs temperature range (130-385 °C), and Reaction 4 between the ANs and HNO₃ range (385-700 °C), the sum of the signals from the ANs and PANs channels before addition of VOC should be equivalent to the $N_2O_5$ signal from the HNO₃ channel (from the $NO_3$ fragment of the $N_2O_5$ dissociating to $NO_2$). For the instrument application shown here, operating at UC Irvine in September 2019, 7% of the modeled $N_2O_5$ produced

based on a model constrained to measured $NO_2$ and $O_3$ is detected in the PNs channel at 150 °C, and 28% in the ANs channel at 385 °C.

## 3.5 Determination of denuder efficiency

The activated carbon denuder was tested for efficient removal of gas phase molecules by flowing gas mixtures of single molecules diluted in zero air through the denuder. Gas mixtures were tested at several concentrations to determine if efficiency

was concentration-dependent. Transmission of the denuder is defined to be the percentage of gas-phase molecules that passed through the denuder and were detected downstream when all should have been removed.

$NO_2$ transmission was tested in 2016 at two relatively low concentrations, to mimic atmospheric conditions, and one higher concentration to mimic chamber conditions. In all cases at this time, greater than 96% of the $NO_2$ was absorbed (Table 1). $NO_2$ concentrations ranged from 26 to 271 ppb. In 2019, $NO_2$ transmission was again tested to assess changes in denuder

performance over time, at 275 ppb. The larger observed $NO_2$ transmission in 2018 suggests a drift in the gas-phase breakthrough over time. Because transmission appears to change over time, we recommend making periodic measurements and updating correction factors accordingly. Denuders can be cleaned by gentle heating and zero air flow.

**Table 1. Effect of inserting a single channel activated carbon denuder in between an $NO_2$ source and the TD-CRDS. Errors were measured for the 2016 measurements and are reported as the standard deviation.**

| Year | [NO₂] (ppb) | NO₂ denuder transmission |
|------|------|------|
| 2016 | 26 | (3.3±0.3) % |
| 2016 | 46 | (3.1±0.2) % |
| 2016 | 271 | (1.96±0.08) % |
| 2019 | 275 | 7.7 % |

The same process was used to determine the transmission of isobutyl nitrate (an alkyl nitrate) in 2016 (Table 2). The outlet of the denuder was connected to both Channel 1 (temporarily at 520°C) and Channel 2 (385°C). The transmission of the denuder was not dependent on the concentration of gas in the original gas mixture, or on which oven was used. This ANs transmission

was also re-tested in 2019, and in this case, no significant change in breakthrough was observed. These measured fractions of gas-phase breakthrough can be used to correct the aerosol measurements made in the denuded channels of the instrument cycle (see corrections discussion below).

**Table 2. Transmission of denuder at three concentrations of isobutyl nitrate (IBN) and one concentration of chamber-generated AN. Transmission is defined as the percentage of gas-phase alkyl nitrate that was passed through the denuder. Errors for the 2016**
**measurements are the standard deviation.**

| Year | AN source | Concentration (ppb) | Transmission through Channel 2 (385°C) |
|------|------|------|------|
| 2016 | IBN | 250 | (13.2±0.3) % |
| 2016 | IBN | 385 | (11.0±0.4) % |
| 2016 | IBN | 800 | (12.8±0.2) % |
| 2019 | Δ-3-carene | 35 | 11.0 % |

A chamber experiment with Δ-carene was performed to generate organic aerosol particles in order to test the aerosol throughput of the denuder. Low $NO_x$ chamber conditions (450 ppb $O_3$, 3 ppb $NO_2$) were used to minimize potential $N_2O_5$ interferences. First, the chamber was hooked directly to the SEMS in order to get a background measure for the number of particles in the

275 bag. Then the chamber mixture was pulled through the TD-CRDS inlet tubing while bypassing the denuder in order to quantify particle losses to the tubing. Finally, the chamber mixture was sampled while flowing through the tubing and the denuder to get the total particle loss through the instrument. The time series of this experiment is shown in Figure 7. In order to quantify the efficiency of the denuder and tubing inlet of the TD-CRDS, the particle volume was averaged over the sampling time under each condition. This assumes particle concentration in the chamber was constant over the course of the entire experiment.

Since the chamber had been running for 24 hours prior to measurements, it is reasonable to assume that all concentrations had reached equilibrium.

A total of 28% of the aerosol particles (assessed by volume) that flow into the instrument were lost to the tubing and the denuder before detection. There does not appear to be any bias toward removing smaller or larger particles. The denuder is responsible for only 10% of total particle loss. This suggests that every deployment of this instrument should carefully consider

and if possible quantify inlet line losses.

### 3.6 Determination of detection limits

The CRDS can be set to zero automatically at regular intervals, which is accomplished by diverting inlet air through an $NO_2$ scrubber. The instrument is typically set to run its 3 minute zero every two hours. The re-zeroing procedure results in small changes to the baseline before and after zeroing events. On ambient measurements, these changes are typically less than 0.5

290 ppb (see Figure S3), and on zero air, typically less than 0.2 ppb, and are sometimes positive and sometimes negative. We determine the standard deviation of four hours of zero measurements (0.16 ppb) to estimate our blank error, $\sigma_{zero}$.

From this observed blank error, the detection limit of the instrument (LOD = $3\sigma$) can be calculated for each channel. The error for the $NO_2$ channel is based only on $\sigma_{zero}$ alone, since no subtraction is required ($3\sigma_{zero} = 3 \times 0.16$ ppb = 0.48 ppb). For all other channels, the error in the subtracted value A - B is calculated as:

$$\sigma_{A-B} = \sqrt{(\sigma_A^2 + \sigma_B^2)} \qquad\qquad\qquad (Eq.\ 3)$$

where $\sigma_A = \sigma_B = 0.16$ ppb are the errors in the pre-subtraction $NO_2$ concentration measurements. Thus, the estimated detection limit for the subtracted channels (ANs, PNs, and $HNO_3$), $3\sigma_{A-B} = 3 \times 0.22$ ppb = 0.66 ppb.

### 3.7 Kinetic modelling of thermal dissociation ovens

Modelling of the ovens can be employed to simulate the dissociation and recombination of the detected species in any oven

design. Pressure- and temperature-dependent rate constants for dissociation (Day et al. 2002) and recombination (JPL Data Evaluation 2015) reactions of PNs, ANs, and $HNO_3$ were used (see Table S1), alongside an assumed (a) step function or (b) linear rate of cooling from the heated to the unheated portions of the oven (see Figure S4). We also included the IUPAC rate constant for a representative RO + $O_2$ ($7.2x10^{-14}$ $e^{-1080/T}$, IUPAC), and OH wall loss rate (calculated to be 46 s$^{-1}$ for these conditions) from Knopf, Pöschl, and Shiraiwa 2015. Based on these rate constants and the assumption that recombination or

wall losses are the only fates for dissociated radicals, we found that the PNs measurement would be the most affected by recombination. We found an expected 10% difference in the amount of PNs recombined by the end of the PNs oven between assuming linear cooling and step function, so the more conservative step function assumption can be used to provide a lower limit concentration.

All ovens were modelled at their setpoint temperature, which is maintained by the thermocouple relay. However, each oven

surely has gradients in temperature along its length, resulting in this average oven temperature measured at its midpoint (see

Figure 1b). As an example, for HNO$_3$ there was very little difference in dissociation based on small changes in oven temperature. HNO$_3$ is 100% dissociated at the end of the oven, so modelling at 30°C hotter than the setpoint temperature just extends the cooling region slightly. This leads to approximately 0.3% less recombination than the setpoint temperature model. The same concept applies to the PNs oven, but the recombination difference is larger (1-2% depending on model molecule) due to a 30°C increase in temperature being a larger percentage of the total temperature.

The PN dissociation and recombination oven model results in Figure S5 show a predicted 55.9% dissociation in the step function model, and 65.4% dissociation in the linear function model with no background NO$_2$ and 10 ppb initial PNs concentration.

In the PNs oven, the only important reactions modelled were the dissociation and recombination of PNs. The background concentration of NO$_2$ was considered and was found to have a significant impact on the recombination rate, especially at high concentrations. Figure S6 shows the percent PNs that remain dissociated at the detector as a function of initial concentrations of PNs and NO$_2$. Two separate types of PN were considered in the modelling, due to their slightly different rates of recombination. Methyl PN gave 58.1% detection at 10 ppb initial concentration and no background NO$_2$, while ethyl PN gave 55.9% detection under the same conditions. Given the relatively small difference in recombination percentages, no effort was made to incorporate both species into the model. Ethyl PN was chosen as the representative species because it was assumed that most PNs being encountered would be two carbons or larger. Including reasonable atmospheric concentrations of OH ($4 \times 10^6$ molecules /cm$^3$) in the model made no difference to the percent recombination of ethyl PNs, and was therefore left out. We note that the PNs measurement will be most affected by recombination, but that this recombination can in principle be corrected for.

In addition to the potential reduction in PNs signal due to recombination reactions, there is the potential for a spurious overestimation of PNs signal due to reactions of thermally dissociated peroxy or peroxy acetyl radicals with ambient NO in the presence of O$_2$, producing additional NO$_2$ (Thieser et al. 2016). This effect will be minimal in chamber simulations of nighttime chemistry, where the mixing ratio of NO is zero, but should be considered in any daytime field deployments.

In the ANs oven, in addition to the major reactions of ANs dissociation and recombination, the reaction RO + O$_2$ is also important. The RO + O$_2$ reaction is extremely fast at high temperatures (see Table S1) like those found in the heated portion of the AN oven, and we assume the reaction to be irreversible. Because O$_2$ is abundant, the reaction negligibly affects O$_2$ concentration. As a result, these assumptions give a model prediction of 100% detection of alkyl nitrates at all initial AN and NO$_2$ concentrations.

In the HNO$_3$ oven, in addition to the dissociation and recombination of HNO$_3$, the loss of OH radical to the walls is significant, competing with recombination. The model assumes that any OH that hits the walls after the heating part of the oven is lost due to reactions with the walls; as a result, recombination is generally less of an effect on the HNO$_3$ measurement. Figure S7 shows example model outputs for the HNO$_3$ oven, predicting the percent dissociation of HNO$_3$ at the point of detection over a large range of initial concentrations for both HNO$_3$ and NO$_2$. As expected, recombination is most important at larger NO$_2$ and HNO$_3$ concentrations; below 50 ppb of each, for this instrument configuration the detection efficiency is above 80%.

### 3.8 Ozone pyrolysis at high temperatures interferes with HNO₃ measurement

One additional reaction that can affect the HNO₃ measurement is the pyrolysis of O₃. At high temperatures, some fraction of O₃ dissociates, releasing atomic O which reacts with NO₂ to form NO + O₂, which results in NO₂ being removed from the final measurement. Therefore, in background conditions of high O₃ concentration, the NO₂ concentrations measured after the HNO₃ oven are biased low and can even cause the [HNO₃] to appear negative upon subtraction. Day et al. 2002 noted that at or above 530 °C, all O₃ will separate into O₂ and O molecules, which will then react with NO₂. This suggests that for this instrument, the pyrolysis of O₃ will result in a lower signal only in Oven 1 (700 °C) due to the high temperatures.

In some experiments from the 2018 SAPHIR NO3ISOP campaign, HNO₃ measurements appeared negative due to lower signals from the hottest channel. Using other available instruments' measurements of O₃ and HNO₃, we determined that approximately 4% of the O₃ signal was converted to this apparent negative HNO₃ signal during one experiment (on 8-Aug-2018). However, this fraction did not appear consistent across experiments, perhaps due to substantial HNO₃ inlet losses, and we did not determine a robust and consistent correction factor for this effect. Given that this does not affect alkyl nitrate measurements, and that there were other measurements of HNO₃ available, we did not pursue this further. But in principle, this is a relatively modest effect that can be corrected for after experimentally determining the efficiency of ozone pyrolysis for a particular inlet oven build.

### 3.9 Data corrections

The above modeled 100% efficiency in detecting ANs is fortunate, since the ANs measurement has thus far been the output of greatest interest from this instrument. Should one wish to use such an instrument for accurate measurements of PNs and HNO₃, this too is possible, but requires the determination of correction factors to account for the recombination in those ovens. Beyond the correction factors for radical recombination in the cooling region after each oven (1), additional corrections that can be applied are: (2) oven-specific denuder breakthrough, based on data such as that shown in Tables 1 and 2, (3) background corrections, to account for any background signal detected in each channel while sampling zero air (this could account for inlet and/or denuder offgassing), (4) subtraction of N₂O₅ interference, as described in section 3.4 above, and (5) correction of the HNO₃ channel for O₃ pyrolysis loss of NO₂, as described in section 3.8 above. The importance of each of these corrections will depend on the nature of the experiments conducted; some example applications are shown below to illustrate this. We have implemented each of these corrections as optional to apply to any raw data collected in our Igor-based data workup routine, which also sorts the data from the various ovens, averages, and subtracts the relevant signals.

## 4 Representative uses of the TD-CRDS

### 4.1 AMS / TD-CRDS aerosol terpene nitrate comparison at CU Boulder chamber

The TD-CRDS was compared to the CU Boulder Jimenez group aerosol mass spectrometer (AMS) during collaborative chamber experiments in Summer 2015, using the data from the denuded ANs channel of the TD-CRDS and the high-resolution AMS organic nitrate (pRONO2) measurement to assess the correlation of these two aerosol-phase organic nitrate measurements. The experiments plotted here are those which showed substantial aerosol nitrate formation using Δ-carene or α-pinene as a VOC precursor and $NO_3$ from an $N_2O_5$ trap, both spanning the nominal range of 10-100 ppb, at varying relative

humidity. The comparison of individual measurements across two weeks of experiments show significant scatter, but an orthogonal distance regression (ODR) fit to the scatterplot of TD-CRDS data vs. AMS data shows a slope of about 0.88-0.94 (depending on intercept treatment), and $R^2=0.73$ (Figure 8).

The AMS organic nitrate concentrations in Figure 9 were calculated by apportioning the total nitrate concentration using the $NO_x^+$ ion ratio ($NO_2^+/NO^+$) method (Farmer et al. 2010), where the relative ratios of organic to inorganic $NO_x^+$ ratios ("ratio-

of-ratios"; Fry et al. 2013) were determined by the average of several dry, unseeded experiments and ammonium nitrate ratios from offline calibrations (3.12 for Δ-carene, 3.78 for α-pinene). The organic-inorganic separation was conducted in order to account for possible $NH_4NO_3$ or particle $HNO_3$ formation as was suggested by substantial shifts in $NO_x^+$ ratios observed during wet, seeded experiments, as has been reported previously (Takeuchi and Ng 2019). Figure S8 shows a comparison of the Figure 8 results to a plot of the TD-CRDS measurements against the AMS total nitrate (unapportioned), the latter resulting in slightly

lower slopes and correlation coefficients.

Previous comparisons between AMS and thermal dissociation-based aerosol organic nitrate instruments have found varying agreement for ambient measurements (Ng et al. 2017). Some of these differences could be due to the fact that the ambient atmosphere contains a mix of diverse products from the oxidation of monoterpenes and isoprene in the presence of other gases; the resulting differing mixes of alkyl nitrate structures could alter the sensitivity of one or both instruments.

### 4.2 Ambient measurements of organonitrates in Portland, OR

During one week in November 2014, the TD-CRDS inlet was situated outside the south end of the Reed College Chemistry building. Simultaneous measurements of $NO_2$, PNs, ANs, and $HNO_3$ were made and one representative day is shown in Figure 9, illustrating typical measurable ambient variability and diurnal cycle.

### 4.3 Chamber measurements of isoprene nitrates at SAPHIR chamber (Jülich, Germany)

The TD-CRDS was also used in the month-long SAPHIR $NO_3$ + isoprene campaign in the summer of 2018. The Simulation of Atmospheric Photochemistry in a Large Reaction Chamber (SAPHIR) is a 270 $m^3$ double-walled Teflon chamber with movable shutters allowing for simulation of both daytime and nighttime chemistry. The experiments were run in batch mode with periodic injections of oxidants and reactants. The reactant concentrations were comparable to real atmospheric

concentrations of $NO_2$, $O_3$, and isoprene. Some experiments were run under humid conditions and some had seed aerosol added to facilitate condensations of gas products into the particle phase.

The low, near-ambient concentrations of reactants used, the small degree of partitioning of isoprene nitrates to the aerosol phase, and the relatively long inlet line required resulted in the aerosol organonitrate products being lower than the limit of detection of the TD-CRDS for the particle-phase ANs monitoring. The gas-phase ANs measurements from the TD-CRDS ranged from sub-ppb up to 16 ppb of organic nitrates, with an observed alkyl nitrate molar yield for $NO_3$ + isoprene of ~ 100% under all explored reaction conditions. In order to determine gas/aerosol partitioning of nitrates, the gas-phase ANs measured by TD-CRDS were compared to AMS organic nitrate aerosol measurement. These results are the subject of a forthcoming paper (Brownwood et al., in preparation, 2020).

## 4.4 Chamber measurements of isoprene nitrates at REC (Portland, OR)

The TD-CRDS was also used for chamber experiments throughout the 2018-2019 academic year at the Reed Environmental Chamber (REC), running experiments similar to those from SAPHIR, but at substantially higher concentrations. These experiments aimed to determine whether gas-particle partitioning coefficients ($K_p$) for the $NO_3$-initiated oxidation of isoprene would be similar in a 0.4 $m^3$ chamber at much higher concentrations to those measured in the 270 $m^3$ SAPHIR chamber at much lower, near-ambient concentrations.

The gas-particle partitioning coefficients calculated in these experiments used the aerosol and total gas + aerosol measurements from the TD-CRDS and a total mass measurement from a Brecthel SEMS (BMI Model 2002). The partitioning coefficients derived from these experiments were $5 \times 10^{-4}$ and $4.4 \times 10^{-3}$ $m^3$ $\mu g^{-1}$, for background aerosol loadings of 230 and 20 $\mu g$ $m^{-3}$, respectively. One of these experiments is shown in Figure 10. The aerosol ($c_{aer}$) and total ANs concentrations, and background aerosol loading ($M_{tot}$) were averaged over the shaded period, aerosol-phase was subtracted from total to obtain $c_{gas}$, from which $K_p$ was determined via Equation 4:

$$K_p = \frac{c_{aero}}{c_{gas}M_{tot}} \qquad\qquad\qquad\qquad \text{(Eq. 4)}$$

The fact that the two experiments at different background aerosol mass loadings ($M_t$) did not give exactly the same $K_p$ value could reflect the uncertainty of these measurements, or that the aerosol partitioning is not perfectly described as absorptive partitioning, or that wall losses change as aerosol loadings change. Most important, the range of $K_p$ measured here fall exactly within the range of values observed over a month of $NO_3$ + isoprene experiments conducted under much lower concentration conditions at the SAPHIR chamber ($5 \times 10^{-4}$ - $6 \times 10^{-3}$ $m^3$ $\mu g^{-1}$, Brownwood et al., in preparation, 2020)

These $K_p$ values were compared to theoretical calculations of $K_p$ predicted by the simplified $p^o_L$ prediction (SIMPOL.1) group contribution method (Pankow and Asher 2008), and we find this average volatility consistent with a tri-functional isoprene nitrates, such as isoprene hydroperoxy nitrate, which has a SIMPOL.1 predicted $K_p$ value of $2.38 \times 10^{-3}$ $m^3$ $\mu g^{-1}$. This shows a promising consistency of equilibrium gas-aerosol partitioning of isoprene nitrate products measured in two dramatically different chambers, and suggests the robustness of the TD-CRDS over a wide range of concentrations.

## 5 Conclusions

Using three custom home-built oven channels, a charcoal denuder, and an automated valve control system, a thermal dissociation cavity ringdown spectrometer (TD-CRDS) was constructed for the speciated measurement of gas- and aerosol-phase organic nitrates, split into the classes $NO_2$, PNs, ANs, and $HNO_3$. This instrument has been successfully demonstrated for measurements on atmospheric simulation chambers operating at a wide range of concentrations and ambient measurements; because of the increased uncertainty in the presence of rapid background changes in $NO_2$ mixing ratio, the TD-CRDS is best suited to chamber studies. Users or developers of similar such instruments are encouraged to consider the several data corrections described herein, which will be more or less important depending on the details of the instrument deployment.

**Acknowledgements**

The authors acknowledge many fruitful conversations with a large number of collaborators, with whom we have enjoyed working as we built and refined this instrument. We thank colleagues at University of Colorado at Boulder: Jose Jimenez, Hyungu Kang, Jason Schroder and Pedro Campuzano-Jost, for sharing AMS data and stimulating discussions during our work together in 2014-2015. Keehan, Day and Fry acknowledge support for this collaboration from NOAA's Climate Program Office's Atmospheric Chemistry, Carbon Cycle, and Climate program Grant # NA13OAR4310070. We also thank colleagues at UC Irvine: Jim Smith, Danielle Draper, and Lia Damm, for stimulating discussions during our work together in 2019-2020. Marsavin, Brownwood, and Fry acknowledge support from the U.S. National Science Foundation (NSF) under Grant # AGS-1762106. We thank Reed College colleagues Ben Ayres and Jay Ewing, and Paul Wooldridge and Ron Cohen (UC Berkeley) for valuable discussions and help with instrument construction, and John Crowley and his students (Max Planck Institute for Chemistry, Mainz, Germany) for many valuable discussions around our work together at Forschungszentrum Jülich in 2018.

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

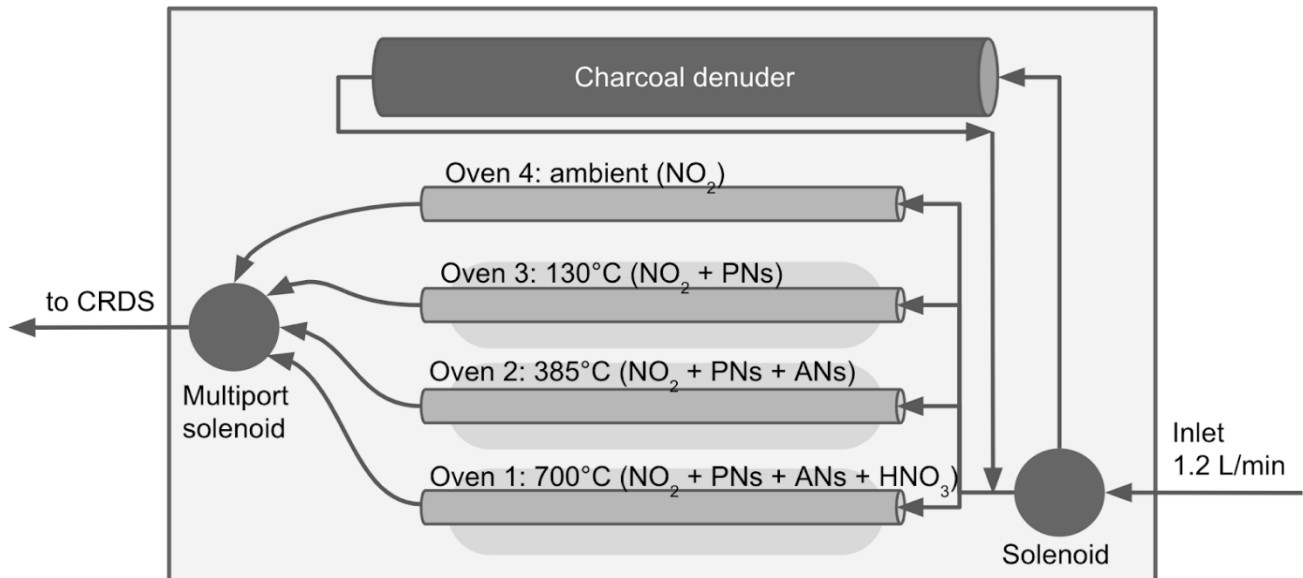

a

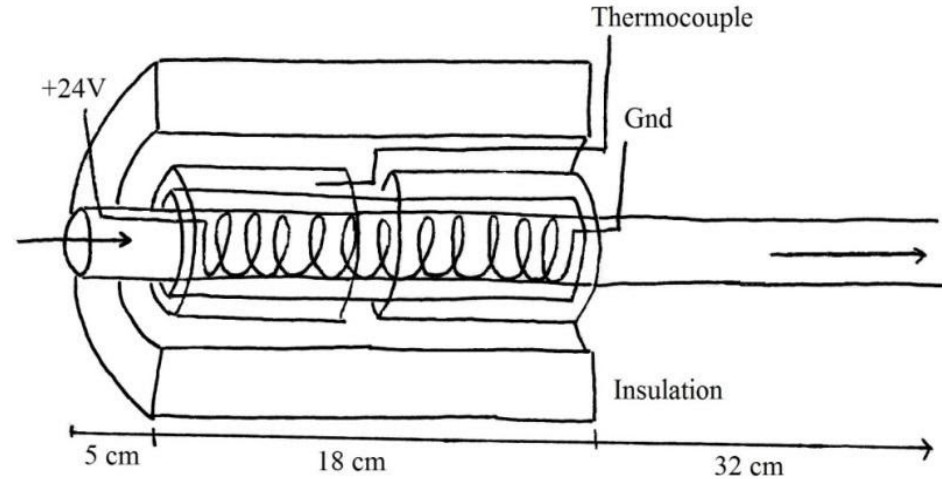

b

Figure 1. (a) Diagram of thermal dissociation inlet flow path. The downstream valve is a Teflon-wetted six-solenoid multiport valve. (b) Oven design. Arrows indicate direction of airflow. A photograph of the inlet box is shown in the Supplemental Information, Figure S1.

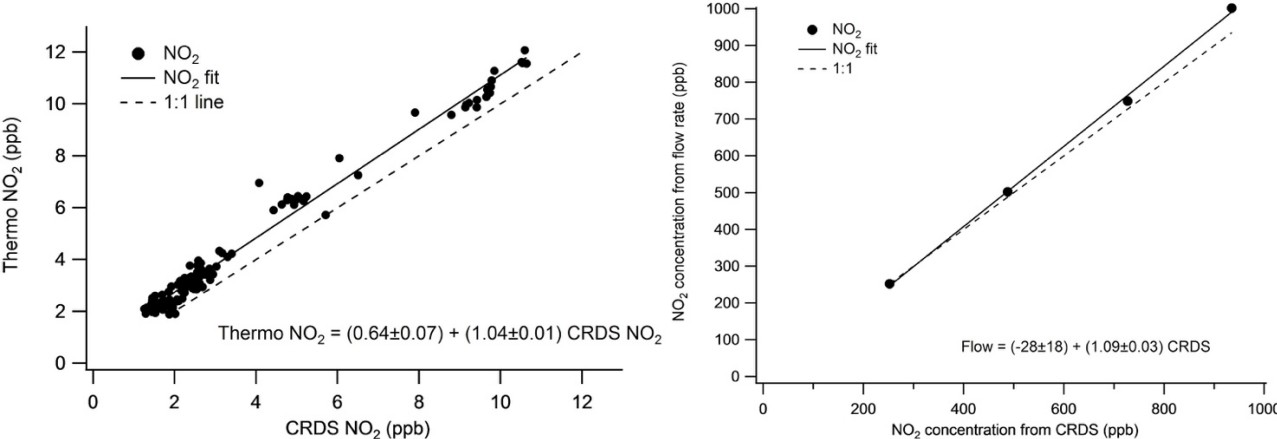

**Figure 2. One full cycle through total and then denuded channels. Points indicated in red are those averaged to obtain the concentrations that are subsequently subtracted. Vertical lines indicate times when the valve switched between channels.**

**Figure 3. (left) Low concentration NO₂ comparison of Los Gatos Research NO₂ cavity ring-down spectrometer to a Thermo chemiluminescent NOₓ box. The dashed line represents a 1:1 relationship. The slope of the fitted line is 1.04 ± 0.01. (right) High concentration NO₂ comparison of LGR cavity ring-down spectrometer to concentrations calculated using flow rates. The dashed line is a 1:1 relationship. The slope of the fitted line is 1.09 ± 0.03.**

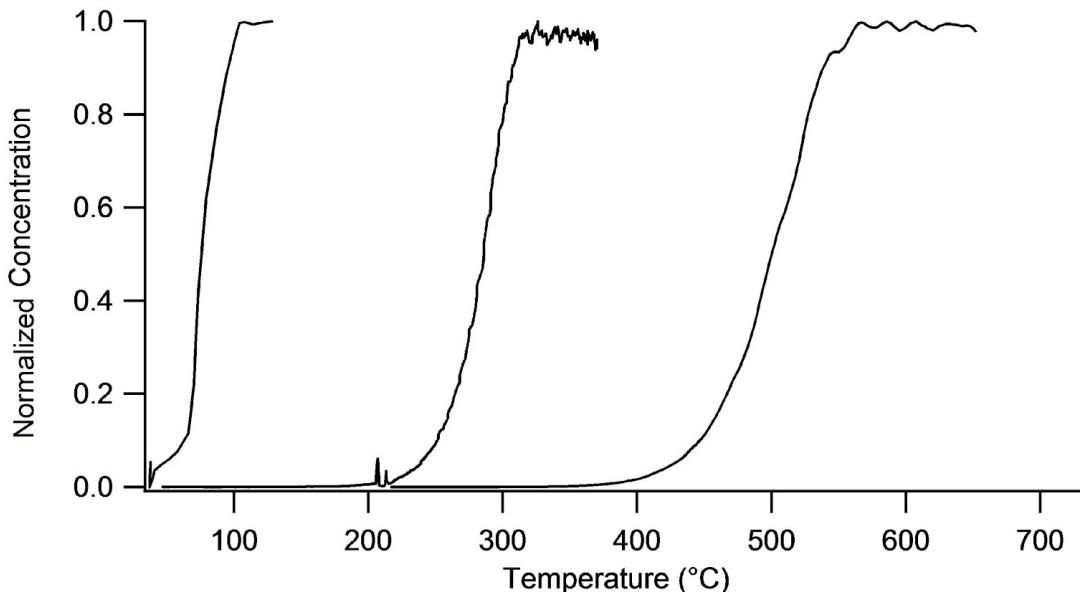

**Figure 4. Temperature ramps on all thermally dissociated species, in order from left to right: chamber-generated PNs, isobutyl nitrate and HNO₃ from pure samples diluted in zero air. The absolute concentrations of PNs, ANs, and HNO₃ were 230, 200, and 3000 ppb, respectively.**

580

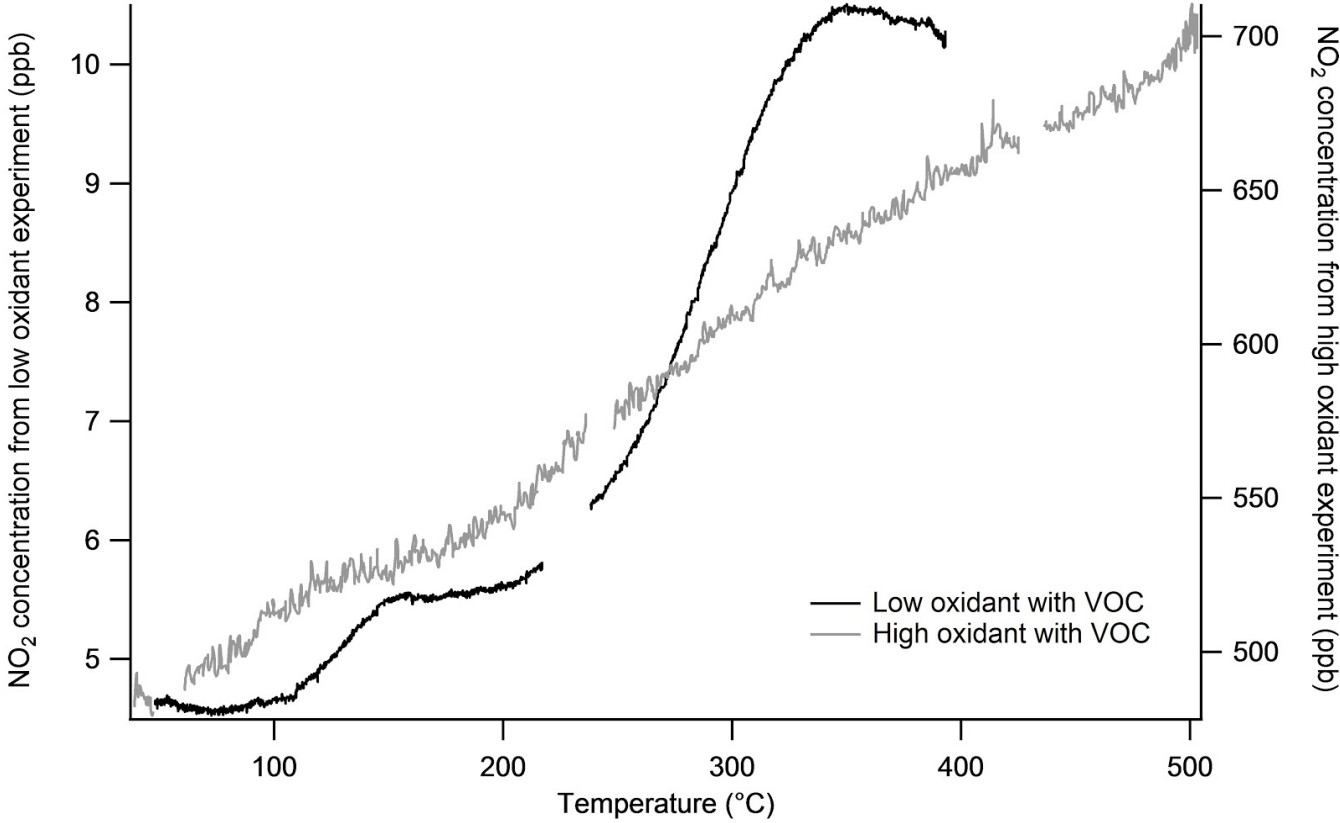

**Figure 5. Comparison of two experiments under different initial NO₂ conditions to change the total N₂O₅ concentration. Both experiments contain 300 ppb Δ-carene. The 'high oxidant' experiment was performed with 650 ppb O₃ and 400 ppb NO₂, which yields substantial N₂O₅ formation. The 'low oxidant' experiment was performed with 335 ppb O₃ and ~3 ppb NO₂, and reveals the clean separation of PNs and ANs by plateaus. There are no distinct plateaus for PNs and ANs in the high oxidant experiment, because they are washed out by the more gradually temperature-dependent dissociation of N₂O₅.**

585

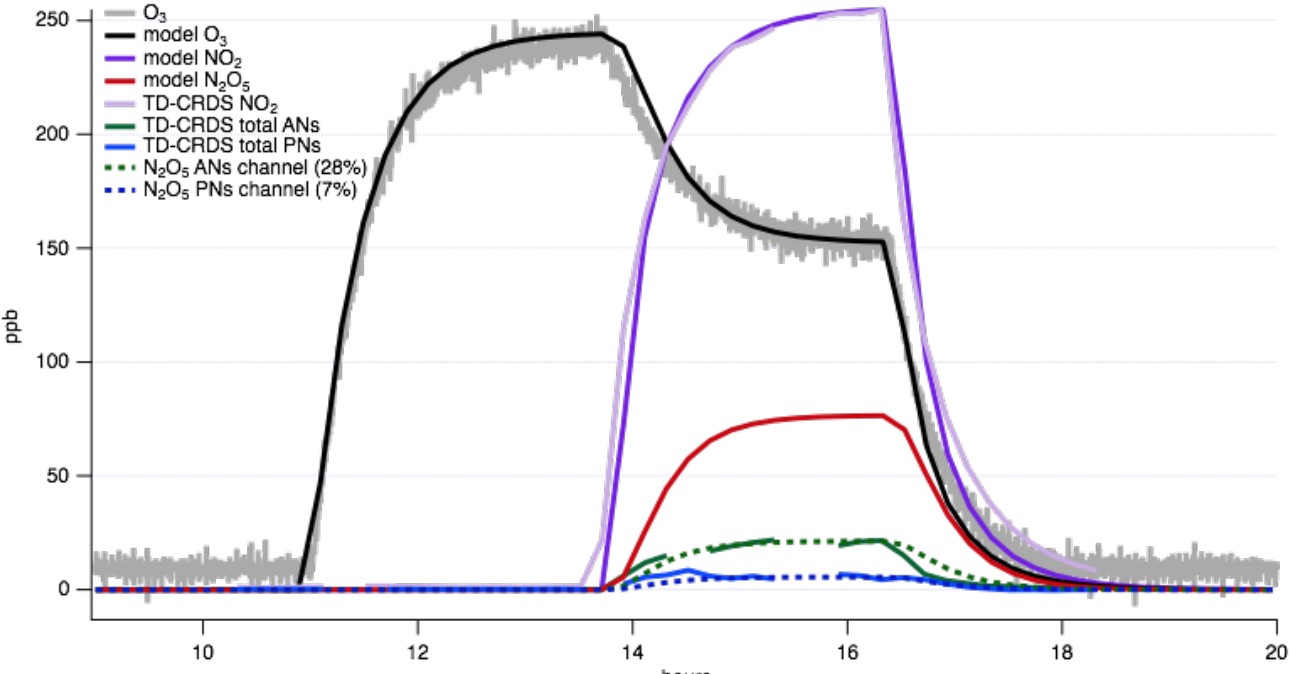

**Figure 6: N₂O₅ contribution to TD-CRDS channels assessed by an oxidant-only chamber experiment. Allowing (measured) NO₂ and O₃ to stabilize sequentially enables prediction of N₂O₅ concentration (red trace) using a kinetics box model, such as KinSim (Peng and Jimenez 2019). Then, the signal in the PNs and ANs channel of the TD-CRDS can be examined during the N₂O₅ rise time, and percentages can be applied to assess the fraction of N₂O₅ that is detected in each channel. For our TD-CRDS, this analysis reveals these percentages are 7% in the PNs channel and 28% in the ANs channel.**

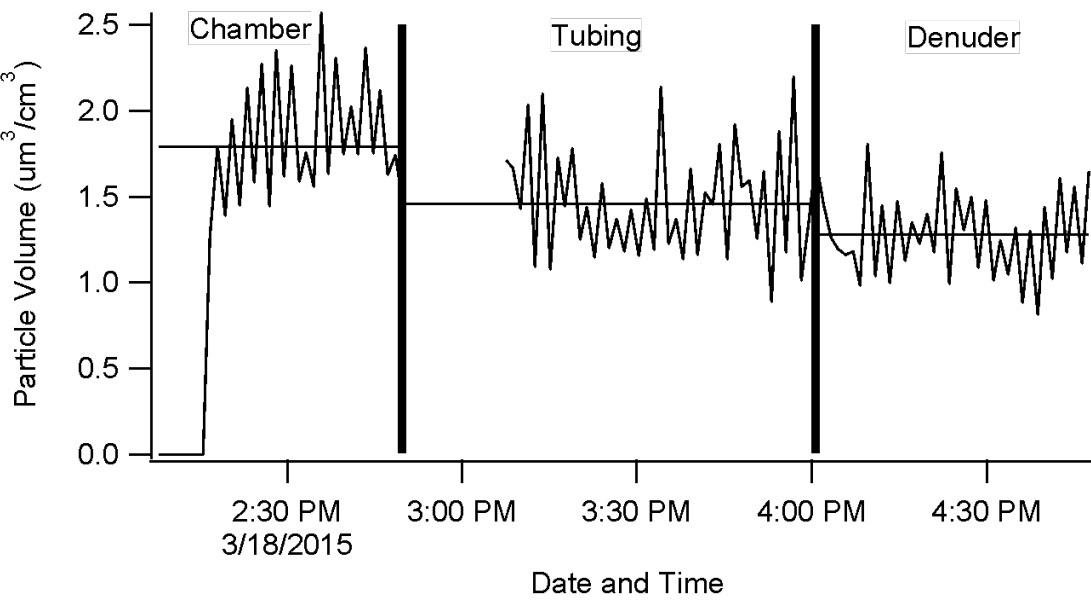

**Figure 7. SEMS-measured particle volume versus time to test denuder efficiency. From 2:15 to 3:00 the SEMS was measuring directly from the chamber. From 3:00 to 4:00 the SEMS was measuring particles from the TD-CRDS tubing (only internal tubing to the inlet system) without the denuder. From 4:00 to 5:00 the SEMS measured through the TD-CRDS tubing and the denuder. The horizontal lines represent the average particle volume over the sampling period. The missing data was due to room air entering the lines while the SEMS was detached from the chamber and reattached to the TD-CRDS inlet.**

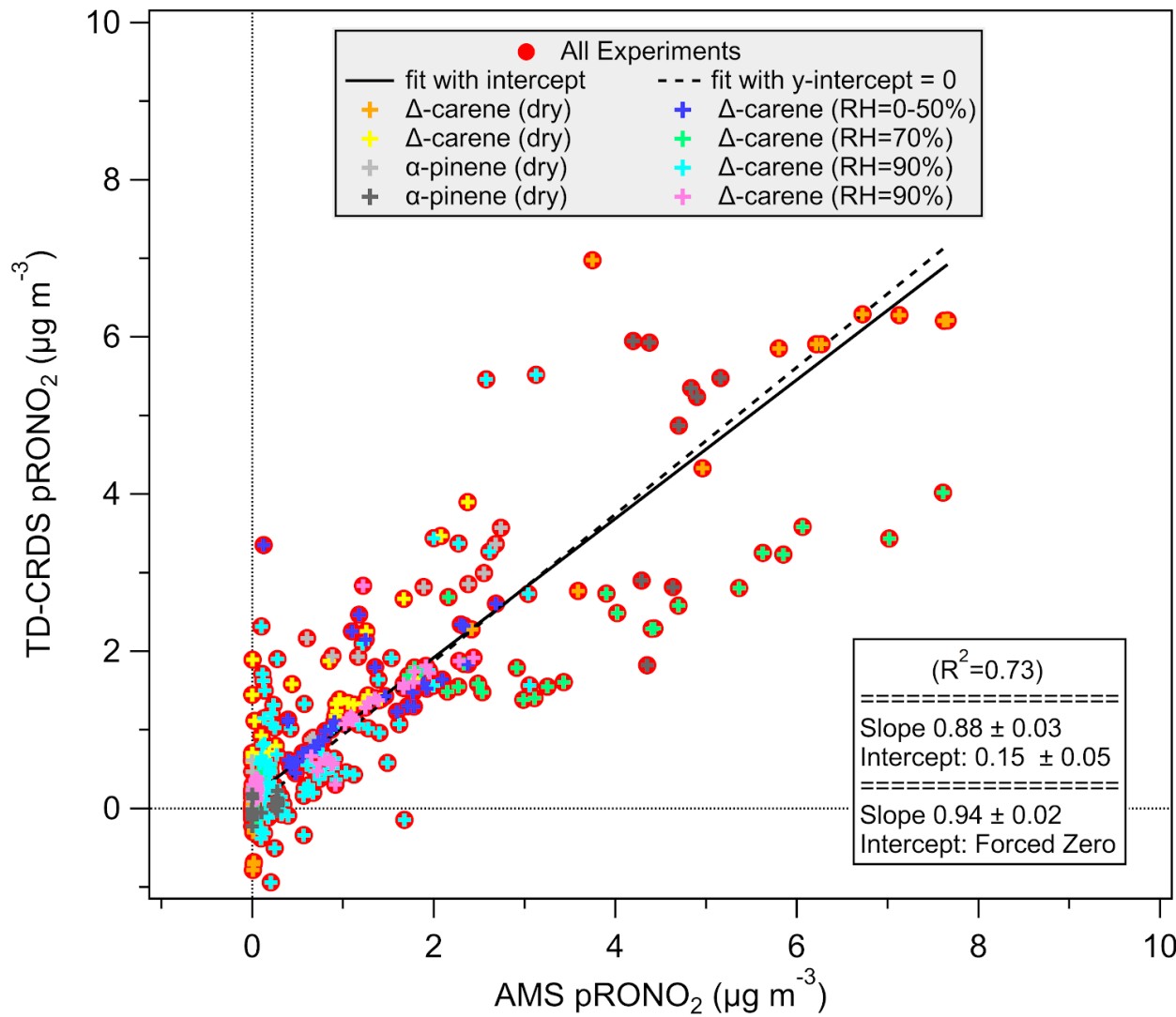

**Figure 8. Background and denuder-breakthrough-corrected aerosol-phase ANs measured by the TD-CRDS, compared to the high-resolution AMS organic nitrate aerosol mass loading. Fits are orthogonal distance regression (ODR).**

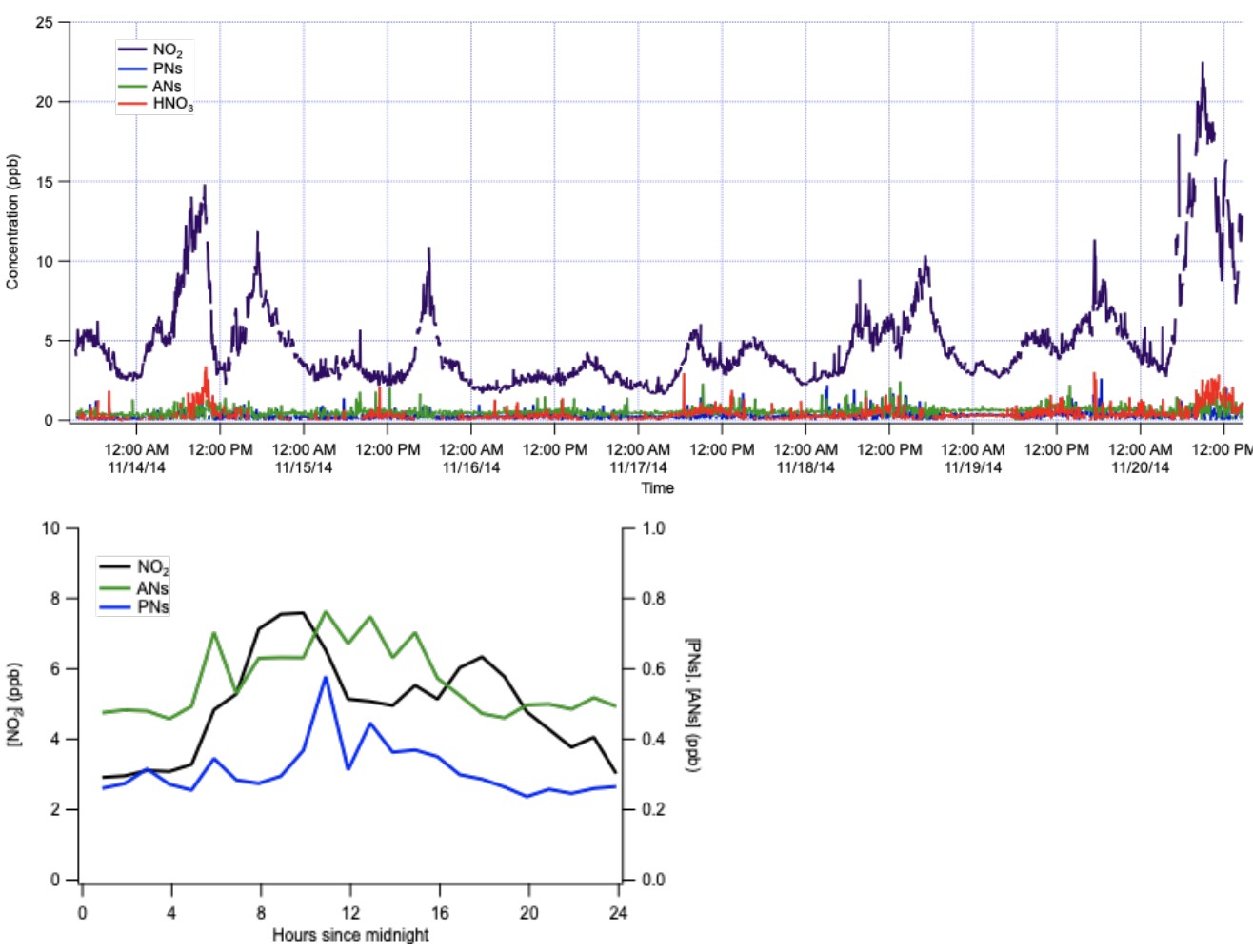

**Figure 9. Representative measurements of NO₂, PN, AN, and HNO₃ concentrations from ambient air in November, 2014 in Portland OR. One week of data is shown to illustrate measurements of typical variability, alongside an average diurnal cycle from this period.**

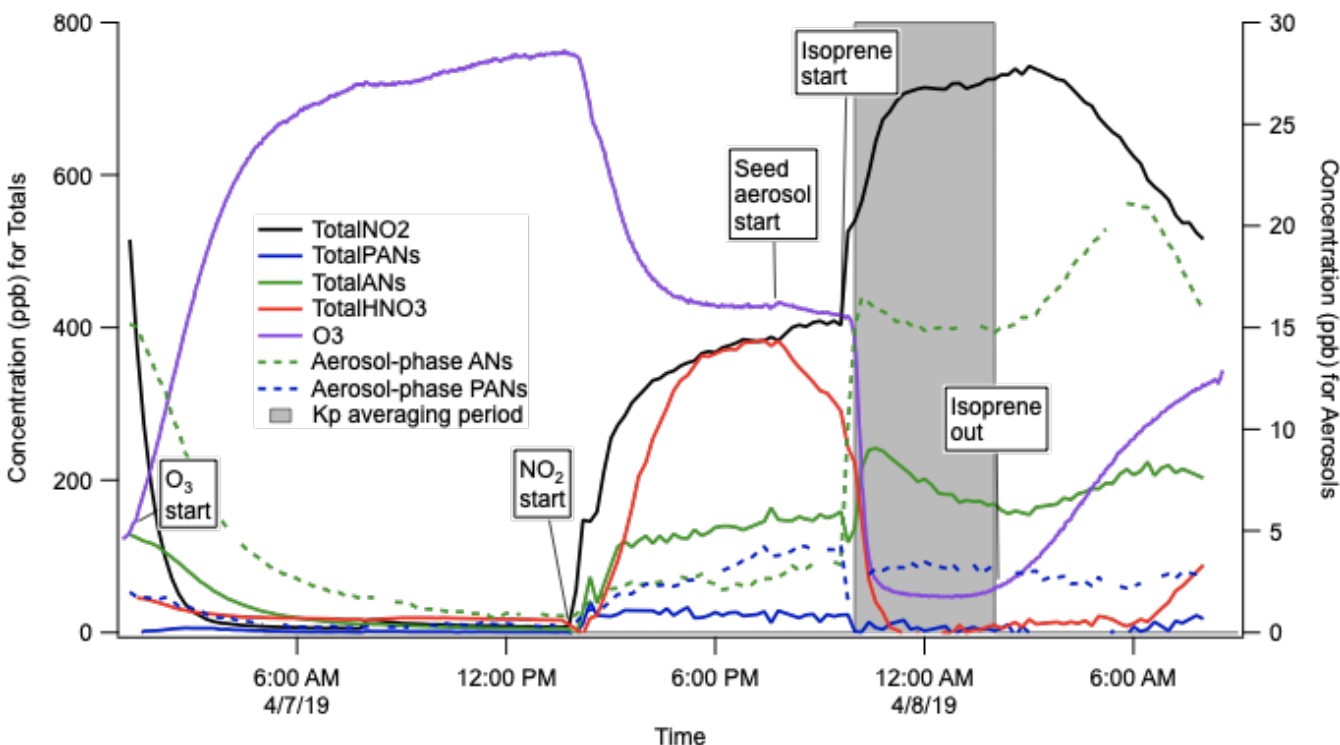

**Figure 10. Example experiment to determine the bulk organonitrate partitioning ($K_p$) from NO₃ + isoprene products in the Reed Environmental Chamber, at 20 µg m⁻³ ammonium sulfate background aerosol. Note that these traces are not corrected for N₂O₅ interferences in the ANs and PANs channel, but N₂O₅ was fully consumed in the period of $K_p$ determination.**

610