# Peer review of "Thermal dissociation cavity ring-down spectrometer (TD-CRDS) for detection of organic nitrates in gas and particle phase"

_Atmospheric Measurement Techniques, 2020_

## Referee Comment (RC1) · Anonymous Referee #1 · 17 Aug 2020

The manuscript presents a TD-CRDS by coupling with a denuder to measure  $NO_2$ , peroxy nitrates (PNs), alkyl nitrates (ANs), and HNO3 in the gas and particle phase. These mentioned NOy species are pyrolyzed under there corresponding temperature windows and produce  $NO_2$ .  $NO_2$  was then measured by a single commercial cavity ring-down  $NO_2$  detector. They showed a feasible way to measure these species in chamber and field studies. They characterized the interference of  $N_2O_5$  under high oxidants condition, and also assessed the interference of the recombination reaction by a model study. This work is valuable, but some comments should be addressed before publication.

**General comments.**

- 1. This work presented the results of the field measurement, but the interference caused by NO in the measurement system had not been considered. The related problems have been studied systematically in the article by Crowley group (e.g., Thieser et al., AMT, 2016; Sobanski et al., AMT, 2016). To make sense, this issue should be discussed.
- 2. Line 127-128, "a liner change," is confused, which is not consistent with Eq. 2. For example, PNs equal to Oven3 minus Oven4, which means the NO2 concentration is not changed during the period of Oven3 and 4. In addition, the time resolution for a cycle is 8 minutes. On this time scale, the NO2 concentration may change due to the emission. A parallel NO2 measurement might helpful in dynamic subtraction.
- 3. Line 150, since the aerosol and gas-phase species, have losses in the denuder and tube, and the aerosol result also affects the following subtraction of gas data, which means the corrections are necessary (the corrections are also important and not easy). The detailed corrections should be added in eq. 2 and well summarized in Sect. 3.9.
- 4. How about the uncertainties of the measurement of these NOy species?
- 5. Before the heated gas and aerosol flowing into the CRDS, do you add a membrane to filter aerosol, if a membrane used, how about the frequency of the filter change, does trapped aerosol have the influence of on the measurement?
- 6. I believe this system is more suitable for chamber study. According to the reported ANs measurement in the previous literatures, the detection capacity of this instrument should be improved for better performance in the field measurement. Figure 9 also showed the ANs below the LOD (0.66 ppbv) in this field study.

**Specific comments.**

- 7. The temperature of the PNs measured in this article is only 130+273 K, is it possible due to the standard samples used in this work is much different with the standard samples applied in previous references, or the measured temperature is not equal to the real temperature in the oven?
- 8. Line 78, delete the redundant "Nitrogen".
- 9. Line 141, how about the time resolution of CRDS-NO2, 1 s, 5 s or 10 s? please clarify it in the manuscript.
- 10. Line 161, "the interference of organic nitrates in the chemiluminescent

measurement," you mean the organic nitrates have the interference of NO2 measurement in CL detector?

- 11. Line 284-285 Knopf et al., 2015 missed in the reference list.
- 12. Line 251, "Error is the standard deviation.", no errors listed here. The column format is not uniform in Table 2
- 13. Line 268, how long is the zero regular interval in general?
- 14. Figure S4, since the linear model labeled as dash line, this figure needs to revise.
- 15. Table S1 C3H70 correct to C3H7O
- 16. Figure S5, "left" and "right" in the caption correct to "top" and "bottom".

Ref

Thieser, J., Schuster, G., Schuladen, J., Phillips, G. J., Reiffs, A., Parchatka, U., Pohler, D., Lelieveld, J., and Crowley, J. N.: A two-channel thermal dissociation cavity ringdown spectrometer for the detection of ambient NO2, RO2NO2 and RONO2, Atmos Meas Tech, 9, 553-576, 10.5194/amt-9-553-2016, 2016.

---

## Referee Comment (RC2) · Anonymous Referee #2 · 18 Aug 2020

Keehan et al. describe the adaptation of a commercial CRDS NO2 instrument for measuring classes of thermally labile nitrates in both the gas and particulate phases. The thermal-decomposition technique for measuring classes of nitrate compounds has been an important tool for constraining concentrations of unknown nitrate species and e.g. NOx / NOy budget closure studies. Typically, these types of measurements have been demonstrated using custom-built NO2 sensors, and it is therefore quite useful to show that a commercial NO2 sensor can also be used to produce sufficient data quality for e.g. laboratory and urban studies.

The paper is well written and thorough and I think deserves publication in AMT after

addressing some suggestions and questions that I outline below. I do think there are some important issues that the authors should address in the revision. These are listed in the specific comments below but I will reiterate them here:

1) Why is the inlet transmission of N2O5 believed to be so low and how do we know that the inlet transmission for other species, e.g. HNO3 or AN, is not also low? NO2 and particulate RONO2 are somewhat validated by comparison to other instruments, but I do not believe that absolute standards of other species are presented. 2) Please carefully check figure 5 and the discussion surrounding the thermal decomposition of N2O5 and NO3, as discussed below. 3) How could a pressure reduction upstream of the heaters change the recombination of thermally decomposed species?
* * *
Line 23: I'm not sure why the word "oxidized" is here.

Line 65-67: Molybdenum catalysts are also widely recognized to convert some other NOy species, not just NO2, into NO, which would cause a significant problem for this work.

Line 67-68: I would say for LIF the limit is laser power not cavity length. A Multipass cell essentially increases laser power in the middle of the cell.

Line 78: remove first word

102-104: I was confused by mentioning LGR CRDS with two flow rates. Recommend clarifying that the two are different instruments and that the second one is for the present work.

106: Metric units please

135: It seems likely the settling time might be significantly reduced by maintaining flow through all channels at all times. Recommend the authors consider this for future deployments, or comment in the manuscript of they know that this would not help.

145: Is there ever any aerosol NO2 detected or could this sampling mode be eliminated?

Section 3.1: Can you say a bit more about how the comparison with the commercial NOx sensor was performed? Was this performed over a short time period by dynamically diluting air from the lab which is expected to be a relatively constant mixture during the experiment? If so, any interference would not be a constant offset but would scale with the dilution. I am actually surprised that the slope is so close to 1, as I expected that the molybdenum converter converted many NOy species. Conversion of Nitric acid seems like another likely positive interference with the CL instrument. It may be that that sensor reports 0.64 ppb even when sampling clean zero air due to a background from the converter. How is the zero for the CRDS determined? Is the laser tuned off of an NO2 resonance or is a periodic zero air sampling period required?

174: In my experience, the certification on those cylinders is not good for more than 1 year and significant loss of NO2 in the cylinders is sometimes observed over longer periods. Perhaps this one is different.

Line 188: "delta-3-carene"

Section 3.3 / Figure 4: How is it known that the observed thermogram from ∼50 – 100C (PNs) is from peroxy nitrates and not from N2O5? Can the authors cite a paper showing that the formation of peroxy nitrates are expected from the reaction of D-3-carene + NO3?

Section 3.4 / Figure 5: I am confused by this figure. The gray line shows much more noise than the black, and when I first looked at it assumed the gray line was for the low oxidant experiment although now see that the caption suggests otherwise. The precision shown on the black line against the left axis seems better than is expected for the stated detection limit of the CRDS instrument. So – can the authors please check that the legend and axes are labeled properly? If they are reversed this would change some of the discussion. Also, for the low oxidant experiment, as shown on the

left axis, the thermogram shows a > 5 ppb range of NO2, while the caption says that 3.2 ppb was used. Could the authors provide a bit of discussion here surrounding what is expected from the experiment, e.g. is it expected that in the low oxidant experiment all of the NO2 would be lost to the NO3 + alkene reaction by the time the air is sampled by the instrument, and so we should expect to see about as much RONO2 as there was initially NO2?

I was somewhat confused by the discussion surrounding the appearance of N2O5 in the thermograms. Initially I thought that the authors were suggesting that N2O5 -> NO3 + NO2 was resulting in the increase in signal > 200C, but later realized they were talking about NO3 -> NO2 + O. I suggest this section starts with a brief discussion of the two-step thermal decomposition of N2O5, and I would not refer to NO3 -> NO2 + O as thermal dissociation of N2O5. Could the authors indicate where N2O5 -> NO3 + NO2 is visible in the thermogram? Also, what effect is there from thermal decomposition of O3 followed by NO3 + O -> NO2 + O2?

I am quite surprised by the very low transmission / detection of N2O5 in the system, as I would not have thought based on the previous similar studies that N2O5 was much more difficult to sample than the other classes of nitrates. The stated detections in the PN and AN channels (7% and 28%) are difficult to reconcile. If N2O5 is completely dissociated in the PN channel, and the conclusion is that only 7% of N2O5 must be transmitted through the inlet, than I would expect at most another 7% of signal from the NO3 decomposition (total of 14% instead of 28%). But still, in the AN channel only a fraction of NO3 is dissociated. Did I miss something here? Is there another study that could be cited that reports low transmission of N2O5 through Teflon tubing?

Section 3.7 / 3.8: The dependence of the inlet heater conversion efficiency and chemistry on the pressure within the heater is not discussed, but may be worth consideration for the authors in the future. My expectation is that if a lower pressure is used within the heater, this would greatly reduce the recombination. Perhaps it is not used that way here because this would require lower pressure within the CRDS and possibly lower

precision. If so, it is a worthwhile point of discussion when considering differences between CRDS and LIF detection of NO2.

Line 284: please include the Knopf et al citation in the Reference list. Also, I presume that the OH loss rate was calculated based on the uptake coefficient stated in that paper using the conditions for this experiment. If so, I suggest the authors state that here because as it is it sounds like the 46 / s number came directly from that paper.

Section 4.1: Were any particulate peroxy nitrates detected using the TD-CRDS instrument? Is it known how those would be classified by AMS?

---

## Author Comment (AC2) · 24 Sep 2020

Please see attached responses to both reviewers. Thank you!

Please also note the supplement to this comment:
https://amt.copernicus.org/preprints/amt-2020-280/amt-2020-280-AC2-supplement.pdf

---

## Author Comment (AC1)

**Response to reviewers for the paper "Thermal dissociation cavity ring-down spectrometer (TD-CRDS) for the detection of organic nitrates in the gas and particle phase," N.I.Keehan, et al.**

We thank the reviewers for their comments on our paper. To guide the review process we have copied the reviewer comments in black text. Our responses are in regular blue font. We have responded to all the referee comments and made alterations to our paper (these are shown in **bold text**).

**Anonymous Referee #1**

Overview

The manuscript presents a TD-CRDS by coupling with a denuder to measure NO2, peroxy nitrates (PNs), alkyl nitrates (ANs), and HNO3 in the gas and particle phase. These mentioned NOy species are pyrolyzed under their corresponding temperature windows and produce NO2. NO2 was then measured by a single commercial cavity ringdown NO2 detector. They showed a feasible way to measure these species in chamber and field studies. They characterized the interference of N2O5 under high oxidants condition, and also assessed the interference of the recombination reaction by a model study. This work is valuable, but some comments should be addressed before publication

**Major comments**

R1.1.This work presented the results of the field measurement, but the interference caused by NO in the measurement system had not been considered. The related problems have been studied systematically in the article by Crowley group (e.g., Thieser et al., AMT, 2016; Sobanski et al., AMT, 2016). To make sense, this issue should be discussed.

Ref Thieser, J., Schuster, G., Schuladen, J., Phillips, G. J., Reiffs, A., Parchatka, U., Pohler, D., Lelieveld, J., and Crowley, J. N.: A two-channel thermal dissociation cavity ringdown spectrometer for the detection of ambient NO2, RO2NO2 and RONO2, Atmos Meas Tech, 9, 553-576, 10.5194/amt-9-553-2016, 2016.

Thank you for pointing this out. We have added the below text in section 3.7 discussing this consideration and referencing Thieser et al.

**"In addition to the potential reduction in PNs signal due to recombination reactions, there is the potential for a spurious overestimation of PNs signal due to reactions of thermally dissociated peroxy or peroxy acetyl radicals with ambient NO in the presence of O$_2$, producing additional NO$_2$ (Thieser et al. 2016). This effect will be minimal in chamber simulations of nighttime chemistry, where the mixing ratio of NO is zero, but should be considered in any daytime field deployments."**

R1.2. Line 127-128, "a linear change," is confused, which is not consistent with Eq. 2. For example, PNs equal to Oven3 minus Oven4, which means the NO2 concentration is not

changed during the period of Oven3 and 4. In addition, the time resolution for a cycle is 8 minutes. On this time scale, the NO2 concentration may change due to the emission. A parallel NO2 measurement might helpful in dynamic subtraction

We agree that the timescale of channel cycling is a substantial limitation of this instrument and concur with the reviewer's suggestion that a parallel NO2 measurement could help in cases where NO2 changes may be faster than changes in NOy. We have revised the text as shown below to clarify the linear change assumption.

"**Any c**oncentration changes **faster than the timescale of the channel cycle** are accounted for by assuming a linear change in each channel between two consecutive samplings of that channel**, and using the interpolated values at the timescale of the measuring channel for subtractions**. This simplifying assumption only holds if the time between channel samplings is relatively short**, and if there are no changes in background $NO_2$ on the timescale of the oven cycling. In situations where rapid $NO_2$ changes are likely, a parallel fast time resolution $NO_2$ measurement could be used to enable corrections for changing $NO_2$ background."**

R1.3. Line 150, since the aerosol and gas-phase species have losses in the denuder and tube, and the aerosol result also affects the following subtraction of gas data, which means the corrections are necessary (the corrections are also important and not easy). The detailed corrections should be added in eq. 2 and well summarized in Sect. 3.9.

We have added a reference to the corrections discussion in Section 3.9 immediately before Eq. 2.

R1.4. How about the uncertainties of the measurement of these NOy species?

Because the uncertainties of NOy measurements are highly situationally dependent (e.g. is background $NO_2$ changing? What type of inlet is required in each experimental situation?), we feel these should be evaluated separately in each deployment of this instrument and do not feel it would be appropriate to assign a single value to them here.

R1.5. Before the heated gas and aerosol flowing into the CRDS, do you add a membrane to filter aerosol, if a membrane used, how about the frequency of the filter change, does trapped aerosol have the influence of on the measurement?

The commercial LGR CRDS instrument does have a teflon membrane filter on its inlet. In situations with high aerosol loading, this filter should be changed regularly to avoid the potential for any additional heterogeneous chemistry on collected aerosol. There is a pressure gauge in the ringdown cell that would provide a warning of a heavily loaded filter.

R1.6. I believe this system is more suitable for chamber study. According to the reported ANs

measurement in the previous literatures, the detection capacity of this instrument should be improved for better performance in the field measurement. Figure 9 also showed the ANs below the LOD (0.66 ppbv) in this field study.

We agree; in chamber studies experiments can be designed to avoid rapid background changes. We have added to the conclusions to underscore this: "This instrument has been successfully demonstrated for measurements on atmospheric simulation chambers operating at a wide range of concentrations and ambient measurements; **because of the increased uncertainty in the presence of rapid background changes in NO$_2$ mixing ratio, the TD-CRDS is best suited to chamber studies.**"

**Technical corrections**

R1.7. The temperature of the PNs measured in this article is only 130+273 K, is it possible due to the standard samples used in this work is much different with the standard samples applied in previous references, or the measured temperature is not equal to the real temperature in the oven?

Yes, it is true that the nominal oven temperature and the 'real' temperature inside the gas flow are not the same (see section 3.3). Complete dissociation was experimentally determined based on chamber generated Δ-carene peroxynitrates.

R1.8. Line 78, delete the redundant "Nitrogen"

We thank the reviewer. The redundant "Nitrogen" has been deleted.

R1.9. Line 141, how about the time resolution of CRDS-NO2, 1 s, 5 s or 10 s? please clarify it in the manuscript.

The time resolution of the CRDS-NO2 is 1 s. The manuscript has been changed to add the following text:

"**Since the CRDS-NO2 takes a measurement every 1 sec, the last three measured points represent 3 seconds of sampling time**"

R1.10. Line 161, "the interference of organic nitrates in the chemiluminescent measurement," you mean the organic nitrates have the interference of NO2 measurement in CL detector?

Yes. See Section 1 where we talk about CL interference due to the molybdenum catalyst. (Wooldridge et al. 2010)

R1.11. Line 284-285 Knopf et al., 2015 missed in the reference list

Thank you, re-added. (Knopf, Pöschl, and Shiraiwa 2015)

R1.12. Line 251, "Error is the standard deviation.", no errors listed here. The column format is not uniform in Table 2

Table 2 and accompanying table caption have been replaced with the following for clarity:

**Table 2. Transmission of denuder at three concentrations of isobutyl nitrate (IBN) and one concentration of chamber-generated AN. Transmission is defined as the percentage of gas-phase alkyl nitrate that was passed through the denuder. Errors for 2016 measurements are the standard deviation.**

| Year | AN Source | Concentration (ppb) | Transmission through Channel 2 (385°C) |
|------|-----------|---------------------|----------------------------------------|
| 2016 | IBN | 250 | (13.2±0.3) % |
| 2016 | IBN | 385 | (11.0±0.4) % |
| 2016 | IBN | 800 | (12.8±0.2) % |
| 2019 | d-carene | 35 | 11.0 % |

Table 1 formatting has been changed to match Table 2, along with clarifying text:

**Table 1. Effect of inserting a single channel activated carbon denuder in between an $NO_2$ source and the TD-CRDS. Errors for the 2016 measurements are the standard deviation.**

| Year | [NO2] (ppb) | NO2 denuder transmission |
|------|-------------|--------------------------|
| 2016 | 26 | (3.3±0.3) % |
| 2016 | 46 | (3.1±0.2) % |
| 2016 | 271 | (1.96±0.08) % |
| 2019 | 275 | 7.7 % |

R1.13. Line 268, how long is the zero regular interval in general?

The following clarifying text has been added to the manuscript:

"**The instrument is typically set to run its 3 minute zero every two hours.**"

R1.14. Figure S4, since the linear model labeled as dash line, this figure needs to revise.

Thank you, we have updated the figure.

R1.15. Table S1 C3H70 correct to C3H7O

Updated.

R1.16. Figure S5, "left" and "right" in the caption correct to "top" and "bottom"

Updated, thank you!

**Anonymous Referee #2**
Keehan et al. describe the adaptation of a commercial CRDS NO2 instrument for measuring classes of thermally labile nitrates in both the gas and particulate phases. The thermal-decomposition technique for measuring classes of nitrate compounds has been an important tool for constraining concentrations of unknown nitrate species and e.g. NOx / NOy budget closure studies. Typically, these types of measurements have been demonstrated using custom-built NO2 sensors, and it is therefore quite useful to show that a commercial NO2 sensor can also be used to produce sufficient data quality for e.g. laboratory and urban studies. The paper is well written and thorough and I think deserves publication in AMT after addressing some suggestions and questions that I outline below. I do think there are some important issues that the authors should address in the revision. These are listed in the specific comments below but I will reiterate them here: 1) Why is the inlet transmission of N2O5 believed to be so low and how do we know that the inlet transmission for other species, e.g. HNO3 or AN, is not also low? NO2 and particulate RONO2 are somewhat validated by comparison to other instruments, but I do not believe that absolute standards of other species are presented. 2) Please carefully check figure 5 and the discussion surrounding the thermal decomposition of N2O5 and NO3, as discussed below. 3) How could a pressure reduction upstream of the heaters change the recombination of thermally decomposed species?

General comments
R2.1) Line 23: I'm not sure why the word "oxidized" is here

This was to indicate that some of the VOCs might already be oxidized, but this is not an essential point. Since it may be confusing, we've removed it.

R2.2) Line 65-67: Molybdenum catalysts are also widely recognized to convert some other NOy species, not just NO2, into NO, which would cause a significant problem for this work.

Precisely this is the benefit of the use of CRDS detection of NO2, rather than chemiluminescence with Mo catalyst, in this work.

R2.3) Line 67-68: I would say for LIF the limit is laser power not cavity length. A Multipass cell essentially increases laser power in the middle of the cell.

We have updated the text to replace the reference to cavity length with laser power.

R2.4) Line 78: remove first word

Done, thank you (see R1.8).

R2.5) 102-104: I was confused by mentioning LGR CRDS with two flow rates. Recommend clarifying that the two are different instruments and that the second one is for the present work.

Different flow rates between our instrument and the instrument described in Paul et al required that for equal residence times, we needed a different length oven. To clarify, we removed "LGR" in front of the CRDS in line 103.

**"A length of 55 cm was calculated from Equation 1 using the Paul et al. CRDS flow rate (q = 2.5 lpm) and oven length (h= 64 cm) in order to give our instrument equal residence times (τ, see Eq. 1). Since the flow rate of the LGR CRDS is significantly smaller (1.2 lpm), the required tube length is shorter than that reported in Paul et al."**

R2.6) 106: Metric units please

The nominal part name was chosen to represent the industry standard size "¼", not a measurement. For clarity the manuscript has been changed to the following:

"**These ovens were attached to nominal ¼ inch (0.635 cm) Teflon tubing with Teflon Swagelok tees and unions. Teflon connectors were chosen over stainless steel to reduce destruction of $NO_2$ by heated steel** (Hargrove and Zhang 2008)**. An oven-length piece of ¼ inch (0.635 cm) Teflon is used as the ambient temperature background $NO_2$ channel, which has a typical temperature of 22 - 24º C inside the inlet box. [...] The denuder is a 45 cm long cylinder of activated charcoal with a ¼ inch (0.635 cm) channel through the center.**"

R2.7) 135: It seems likely the settling time might be significantly reduced by maintaining flow through all channels at all times. Recommend the authors consider this for future deployments, or comment in the manuscript if they know that this would not help.

The following text was added to section 2 to address the possibility of a continuous flow setup:

**"(We note that maintaining a constant flow through each of the channels at all times would help to reduce the stabilization time in the CRDS, leading to a reduced time resolution. Because the CRDS has its own internal pump to draw air into the CRDS cell, a secondary pump would be required to maintain constant air flow through the non-sampling channels. Such a modification could help make this instrument more viable for high time resolution ambient measurements.)"**

R2.8) 145: Is there ever any aerosol NO2 detected or could this sampling mode be eliminated?

We have found "NO2 aerosol mode" useful in diagnosing instrument contamination problems, interferences not yet accounted for, and false values caused by rapid changes to an unstable system. It isn't totally necessary, but it can be valuable for data analysis post-experiment. We sometimes see signal in the "NO2 aerosol" channel, but it is usually indicative of other instrument issues, not "real" NO2 aerosol.

The following addition was made to the manuscript:
"**While there is not expected to be any signal in the NO2 aerosol channel, the channel has proven useful for diagnosing contamination problems, interferences not yet accounted for, and false values caused by rapid changes.**"

R2.9) Section 3.1: Can you say a bit more about how the comparison with the commercial NOx sensor was performed? Was this performed over a short time period by dynamically diluting air from the lab which is expected to be a relatively constant mixture during the experiment? If so, any interference would not be a constant offset but would scale with the dilution. I am actually surprised that the slope is so close to 1, as I expected that the molybdenum converter converted many NOy species. Conversion of Nitric acid seems like another likely positive interference with the CL instrument. It may be that that sensor reports 0.64 ppb even when sampling clean zero air due to a background from the converter. How is the zero for the CRDS determined? Is the laser tuned off of an NO2 resonance or is a periodic zero air sampling period required?

Yes. This was performed using dilutions of lab air in zero air over a short period of time. The fact that the slope is so close to 1 is likely due to ambient concentrations of $NO_y$ being low compared to $NO_2$, explaining why the interference does not significantly scale with dilution. The urban location of the lab would support the relatively low levels of $NO_y$ compared to $NO_2$. The 4% difference in slope could be from dilution effects of the $NO_y$ being detected as $NO_2$.

We agree that a converter background could account for some or all of the zero offset in the Thermo chemiluminescence instrument. That's why this instrument is not designed around a Thermo CL.

The LGR has an internal $NO_2$ scrubber that it uses to generate zero air for the zero measurement, described on line 280.

The text has been amended:

"**Since this experiment was performed using dilutions of zero air, any interference from NOy species in the CL detector would also be expected to scale with dilution. The urban location of the lab would support the relatively low levels of NOy compared to NO2, explaining the very small 4% difference in slope between the two detectors.**"

and

"**The intercept offset of the low concentration experiment is 0.64 ppb, which may be attributable to the interference of organic nitrates in the chemiluminescence measurement, or a slight zero offset in the chemiluminescence detector**"

R2.10) 174: In my experience, the certification on those cylinders is not good for more than 1 year and significant loss of NO2 in the cylinders is sometimes observed over longer periods. Perhaps this one is different.

True, this is an older cylinder, but it's the only one we had available. This concern is valid.

R2.11) Line 188: "delta-3-carene"

The line was corrected to read:

"**NO3 + Δ-3-carene mixture**"

R2.12) Section 3.3 / Figure 4: How is it known that the observed thermogram from ~50 – 100C (PNs) is from peroxy nitrates and not from N2O5? Can the authors cite a paper showing that the formation of peroxy nitrates are expected from the reaction of D-3- carene + NO3?

The concentration of reactants is chosen such that all $NO_3$, and hence $N_2O_5$, is consumed in this experiment. Further empirical evidence of the lack of $N_2O_5$ is that it does not plateau at 130 C, as do PNs (see figure 5). Unlike ANs and PNs that dissociate in a plateau at a specific temperature, N2O5 seems to partially dissociate over a large range of temperatures. (see also Womack et al 2017)

R2.13) Section 3.4 / Figure 5: I am confused by this figure. The gray line shows much more noise than the black, and when I first looked at it assumed the gray line was for the low oxidant experiment although now see that the caption suggests otherwise.

The precision shown on the black line against the left axis seems better than is expected for the stated detection limit of the CRDS instrument. So – can the authors please check that the legend and axes are labeled properly? If they are reversed this would change some of the discussion.

The legend is accurate. Likely the noise of the high oxidant experiment is very large because we were approaching the upper limits of the CRDS, where ringdown times are so rapid that their fit is more poorly defined, resulting in an increase in noise.

Also, for the low oxidant experiment, as shown on the left axis, the thermogram shows a > 5 ppb range of NO2, while the caption says that 3.2 ppb was used. Could the authors provide a bit of discussion here surrounding what is expected from the experiment, e.g. is it expected that in the low oxidant experiment all of the NO2 would be lost to the NO3 + alkene reaction by the time the air is sampled by the instrument, and so we should expect to see about as much RONO2 as there was initially NO2?

The 3.2 ppb was an estimate from the lowest possible $NO_2$ concentration possible from our 514 ppm $NO_2$ cylinder calculated by flow rate, by dilution in zero air. The mixing ratio listed is thus spuriously precise, and there is likely some zero offset, which we do not routinely correct for in thermograms. To avoid confusion, we change the reported NO2 mixing ratio to be clearer about how it is an approximate amount (~ 3 ppb instead of 3.2 ppb).

I was somewhat confused by the discussion surrounding the appearance of N2O5 in the thermograms. Initially I thought that the authors were suggesting that N2O5 -> NO3 + NO2 was resulting in the increase in signal > 200C, but later realized they were talking about NO3 -> NO2 + O. I suggest this section starts with a brief discussion of the two-step thermal decomposition of N2O5, and I would not refer to NO3 -> NO2 + O as thermal dissociation of N2O5. Could the authors indicate where N2O5 -> NO3 + NO2 is visible in the thermogram? Also, what effect is there from thermal decomposition of O3 followed by NO3 + O -> NO2 + O2?

N2O5  to NO2 and NO3 is not a nice plateau on the thermogram, thus the problem. This first dissociation step seems to occur split across the PNs and ANs ovens. The second dissociation step (NO3 -> NO2 + O) may also contribute in the ANs channel, and the rest in the HNO3 channel. The suggestion to clarify this at the beginning of this section is good; we added earlier reference to reactions R3 & R4

I am quite surprised by the very low transmission / detection of N2O5 in the system, as I would not have thought based on the previous similar studies that N2O5 was much more difficult to sample than the other classes of nitrates. The stated detections in the PN and AN channels (7% and 28%) are difficult to reconcile. If N2O5 is completely dissociated in the PN channel, and the conclusion is that only 7% of N2O5 must be transmitted through the inlet, than I would expect at most another 7% of signal from the NO3 decomposition (total of 14% instead of 28%). But still, in the AN channel only a fraction of NO3 is dissociated. Did I miss something here? Is there another study that could be cited that reports low transmission of N2O5 through Teflon tubing?

Inlet transmission is only part of the problem, dissociation of N2O5 spread across multiple temperatures also complicates detection. We interpret this as arising because the

recombination to N2O5 is rapid, and thus some NO2 remains bound up in this reservoir until very high temperatures. The first dissociation does not occur to completion in the PNs channel, so the lower percentage there (7%) does not mean only another 7% would dissociate. In fact, it seems that the 7% and 28% are both mainly the first dissociation of N2O5, with the second dissociation (NO3) occurring only at the highest temperature oven.

This text was added to section 3.4 to clarify this: "We note that due to its high reactivity and wall losses (especially the $NO_3$ fragment), **as well as the likelihood that some $N_2O_5$ remained incompletely dissociated even at the ANs oven temperature**, the total $N_2O_5$ detection is substantially less than 100% of the $N_2O_5$ concentration present in the chamber. **We also emphasize that these percentages are specific to the configuration used in this characterization experiment: from the chamber containing the modeled $N_2O_5$ concentration used to determine these interference percentages, a 2-m Teflon inlet line led to the TD-CRDS instrument."**

R2.14) Section 3.7 / 3.8: The dependence of the inlet heater conversion efficiency and chemistry on the pressure within the heater is not discussed, but may be worth consideration for the authors in the future. My expectation is that if a lower pressure is used within the heater, this would greatly reduce the recombination. Perhaps it is not used that way here because this would require lower pressure within the CRDS and possibly lower precision. If so, it is a worthwhile point of discussion when considering differences between CRDS and LIF detection of NO2.

We agree that lower pressure would absolutely help reduce recombination, but would not be possible using this commercial LGR CRDS back end. We added a mention of the advantage of lower cell pressure to reduce recombination to the discussion of comparison to LIF in the introduction:

 "LIF can be tuned to a specific spectroscopic transition like CRDS,  **and can be run at lower cell pressures that reduce recombination (see section 3.7 below)**, but …"

R2.15) Line 284: please include the Knopf et al citation in the Reference list.

Thank you, added, see R1.11.

Also, I presume that the OH loss rate was calculated based on the uptake coefficient stated in that paper using the conditions for this experiment. If so, I suggest the authors state that here because as it is it sounds like the 46 / s number came directly from that paper.

Text was changed to read:

"**[...] and OH wall loss rate (calculated to be 46 s-1 for these conditions) from Knopf, Pöschl, and Shiraiwa 2015.**"

R2.16) Section 4.1: Were any particulate peroxy nitrates detected using the TD-CRDS instrument? Is it known how those would be classified by AMS?

We did not operate at conditions that produced substantial peroxy nitrates for this comparison, and we do not know if these would also appear as pRONO2 to the AMS.